# Vascular dysfunction is at the onset of oxaliplatin-induced peripheral neuropathy symptoms in mice

Sonia Taïb[1],*, Juliette Durand[1],*, Vianney Dehais[1],†, Anne-Cécile Boulay[1],†, Sabrina Martin[1], Corinne Blugeon[2], Laurent Jourdren[2], Rémi Freydier[3], Martine Cohen-Salmon[1], Jamilé Hazan[1], Isabelle Brunet[1]

**Oxaliplatin-induced peripheral neuropathy (OIPN) is an adverse side effect of this chemotherapy used for gastrointestinal cancers. The continuous pain experienced by OIPN patients often results in the reduction or discontinuation of chemotherapy, thereby affecting patient survival. Several pathogenic mechanisms involving sensory neurons were shown to participate in the occurrence of OIPN symptoms. However, the dysfunction of the blood-nerve barrier as a source of nerve alteration had not been thoroughly explored. To characterise the vascular contribution to OIPN symptoms, we undertook two comparative transcriptomic analyses of mouse purified brain and sciatic nerve blood vessels (BVs) and nerve BVs after oxaliplatin or control administration. These analyses reveal distinct molecular landscapes between brain and nerve BVs and the up-regulation of transcripts involved in vascular contraction after oxaliplatin treatment. Anatomical examination of the nerve yet shows the preservation of BV architecture in the acute OIPN mouse model, although treated mice exhibit both neuropathic symptoms and enhanced vasoconstriction reflected by hypoxia. Moreover, vasodilators significantly reduce oxaliplatin-induced neuropathic symptoms and endoneurial hypoxia, establishing the key involvement of nerve blood flow in OIPN.**

## Introduction

Chemotherapy-induced peripheral neuropathy (CIPN) is a detrimental side effect of cancer treatment using neurotoxic chemotherapeutic agents including platinum compounds, taxanes, vinca alkaloids, and proteasome inhibitors (Dorsey et al, 2019). CIPN sensory symptoms include acute burning or stabbing pain, tingling and numbness of the hands and feet, and increased sensitivity to cold temperatures (Müller et al, 2021). These sensory symptoms tend to be more prominent than motor or autonomic dysfunction, although their association is observed in CIPN patients (Staff et al, 2017). The patients' quality of life is significantly reduced because of continuous allodynia and paresthesia, which may result in the reduction or discontinuation of chemotherapy and thereby impact both treatment efficacy and patient survival (Cavaletti & Marmiroli, 2010; Burgess et al, 2021). Moreover, CIPN may be long-lasting or permanent even after the cessation of cancer treatment, whereas no efficient cure capable of preventing its occurrence or improving its long-term course has been developed to date (Cavaletti & Marmiroli, 2015).

Chemotherapeutic agents lead to neurotoxicity through a variety of molecular and cellular mechanisms that may predominantly affect diverse neuronal populations but ultimately trigger dying-back axonopathy and to a lesser extent neuronopathy (Park et al, 2023). The pathogenic mechanisms involved in CIPN depend on the drug mode of action and include distinct subcellular processes such as nuclear or mitochondrial DNA damage, disruption of microtubule dynamics, altered mitochondrial function, and perturbed ion channel activity (Park et al, 2023). Among chemotherapeutic agents simultaneously acting at different subcellular levels, platinum-based drugs are the most neurotoxic, with oxaliplatin causing the highest prevalence of CIPN (70–85% of patients; Argyriou et al, 2013; Burgess et al, 2021). Oxaliplatin is commonly used for the treatment of gastrointestinal solid tumours and particularly in first-line chemotherapy for advanced colorectal cancer (Grothey & Venook, 2018), the second most deadly cancer worldwide.

Although the molecular and cellular mechanisms leading to chemo-induced neuropathic pain are far from being deciphered, the neuronal component of CIPN has been extensively explored in an attempt to identify therapeutic solutions to alleviate symptoms in cancer patients, with very limited therapeutic success. However, the impairment of the blood-nerve barrier (BNB) has recently emerged as a key mechanism triggering neuropathic phenotype initiation in several preclinical neuropathic pain models of

[1]Center for Interdisciplinary Research in Biology (CIRB), Collège de France, CNRS UMR 7241, INSERM U1050, Université PSL, Paris, France   [2]Genomic Facility, Institut de Biologie de l'ENS (IBENS), École Normale Supérieure, CNRS UMR 8197, INSERM U1024, Université PSL, Paris, France   [3]HydroSciences Montpellier, Université de Montpellier, CNRS, IRD, Montpellier, France

Correspondence: isabelle.brunet@college-de-france.fr
*Sonia Taïb and Juliette Durand contributed equally to this work
†Vianney Dehais and Anne-Cécile Boulay contributed equally to this work

traumatic nerve injury (Reinhold et al, 2023), suggesting a possible role for the nervous vasculature. Indeed, peripheral nerves are highly vascularized from embryonic day E16 (Taïb et al, 2022) and critically rely on blood supply to ensure axon structure and function. Furthermore, in the development of neuropathic pain associated with diabetic peripheral neuropathy, alteration of blood flow patterns through the *vasa nervorum* has been implicated (Archer et al, 1984; Nukada, 2014). In addition, reduced blood flow perfusing peripheral nerves was also suggested to be associated with allodynia in mouse models of CIPN (Gauchan et al, 2009; Ogihara et al, 2019). Given that oxaliplatin is injected intravenously, a key vascular component giving rise to nerve dysfunction and neuropathic symptoms remains to be thoroughly characterised in oxaliplatin-induced peripheral neuropathy (OIPN).

This prompted us to first compare vascular sciatic nerve specificities and properties with respect to the brain vasculature. We also explored whether and how the blood vessels of sciatic nerve from animals injected with oxaliplatin were affected compared with those from sham control animals. To this end, we achieved a comprehensive study at molecular, anatomical, and functional levels. We here postulate that the vascular integrity within peripheral nerves could be fine-tuned by a balance of vascular permeability and contractility, maintaining peripheral nerve homeostasis and subjected to perturbations in OIPN condition.

# Results

## An optimised method to isolate intact blood vessels from mouse brain and sciatic nerve

Given that most anti-cancer drugs, including oxaliplatin, are unable to cross the blood-brain barrier (BBB) (Achar et al, 2021), when they specifically target peripheral neurons, we first aimed at comparing the vascular properties of the *vasa nervorum* and the BBB. To this end, we optimised and adapted to peripheral nerves a reported method designed to purify mouse brain vessels (Fig 1A). This protocol includes a partial basal lamina enzymatic digestion to remove non-vascular (glia and neuronal) associated elements (Boulay et al, 2017, 2019). The purified blood vessels (BV) from both adult mouse sciatic nerves (snBV) and brain (brBV) showed a physiological expression of the endothelial cell marker *Pecam-1*/CD31 as shown in qRT-PCR (Fig 1B–D) where it appeared as expected enriched in the BV compared with the entire tissue (Fig 1B) and by snBV immunostaining (Fig 1E, left panel). The isolated snBV showed a continuous staining with isolectin B4 labeling the basal lamina surrounding the endothelial cell layer (Fig 1E, middle panel), which further suggested the BV integrity. This optimised protocol allowed the vascular smooth muscle cells (vSMC) to remain attached to the isolated snBV as established by both the expression of *Sm22a* transcript in qRT-PCR experiments (Fig 1C and D) and immunostaining of alpha-smooth muscle actin (SMA, Fig 1E, right panel). Finally, the absence of neuronal (*Tuj1*) and glial (*Gfap*, *S100b*, *Mpz*, and *Prx*) marker transcripts in snBV (Fig 1C) and brBV (Fig 1D) indicated that our purified BV preparations were devoid of any neuronal or glial contaminations.

## Molecular profiling of purified sciatic nerve and brain blood vessels

To determine the molecular identity of the *vasa nervorum* and compare it with one of the most selective barriers of the body, the BBB, we undertook a comparative transcriptomic analysis between isolated adult mouse snBV and brBV using bulk RNA sequencing. Notably, the overall analysis using an adjusted $P$-value < 0.001 and $\log_2$ fold change < −1.5 or $\log_2$ fold change > 1.5 resulted in the identification of 2,524 commonly expressed transcripts, whereas specific genes were considered when, in addition to those parameters, a given gene was not detected above 500 reads in brain (snBV specific) or in the nerve (brBV specific). It reveals 469 mRNAs highly enriched in snBV (snBV specific) and 502 mRNAs highly enriched in brBV (brBV specific) mRNAs (Fig 2A). Assessing the mean expression level of differentially expressed genes (DEGs) in snBV or brBV showed an enriched gene set in each compartment (Fig 2B). To better characterise the molecular properties of the *vasa nervorum*, which would necessarily underlie its contribution to nerve physiology and physiopathology, we selected two snBV-highly enriched DEGs, *Fabp4* and *Plvap*, and four genes commonly expressed in both datasets and known for their key function at the BBB, *Tjp1*/ZO-1, *Abcg2*/BCRP and *Abcb1a*, *Abcb1b*/ABCB1 (Fig 2B). For the snBV-enriched DEGs, *Fabp4* and *Plvap* were chosen according to their distinct levels of expression (Fig 2B) and respective role in active (*Fabp4* encoding Fatty Acid Binding Protein 4 [FABP4]; Elmasri et al, 2009; Jabs et al, 2018) or passive (*Plvap* encoding Plasmalemma vesicle-associated protein -[PLVAP]; Chang et al, 2023) transport across the vessel wall. In addition, the structural and functional integrity of the BBB is directly controlled by *Tjp1*-encoding ZO-1-positive tight junctions between brain endothelial cells (Huber et al, 2001). Likewise, *Abcg2*-encoding BCRP and *Abcb1*-encoding ABCB1 are critical transporters at the BBB that restrict the brain permeability to its physiological substrates and anti-cancer drugs in vivo (Cisternino et al, 2004). Addressing the expression and indirectly the potential contribution of these selected genes to the blood-nerve barrier was thus of high interest.

The *vasa nervorum* is composed of different anatomical compartments with respect to the position of myelinated axon fascicles (see Fig 2C; Richner et al, 2018): (i) the vessels located within each axon fascicle belong to the endoneurium, (ii) those localized at the periphery of axon fascicles are from the perineurium, whereas (iii) those running along the external side of the nerve belong to the epineurium. The epineurial, perineurial, and endoneurial BVs run longitudinally along the nerve with perpendicular communicating branches, which altogether provides an adequate blood supply for peripheral nerve homeostasis and function. Aiming at exploring the expression pattern of our four candidates in the *vasa nervorum*, we carried out co-immunostaining experiments on sciatic nerve longitudinal sections or whole mount (Fig 2D and E). If *Tjp1*/ZO-1 colocalised with endothelial cell marker *Pecam-1*/CD31 at spots along both endoneurial and epineurial snBV (Fig 2D, upper panels), *Abcg2*/BCRP was only found colocalised with *Pecam-1*/CD31 along endoneurial snBV (Fig 2D, middle panels). Reciprocally, the peripheral nerve-enriched PLVAP selectively colocalised with endothelial cells along epineurial snBV but was almost absent from endoneurial vessels (Fig 2D, lower panels). Moreover, the highly enriched FABP4 was indeed present all along the snBV without strong colocalisation

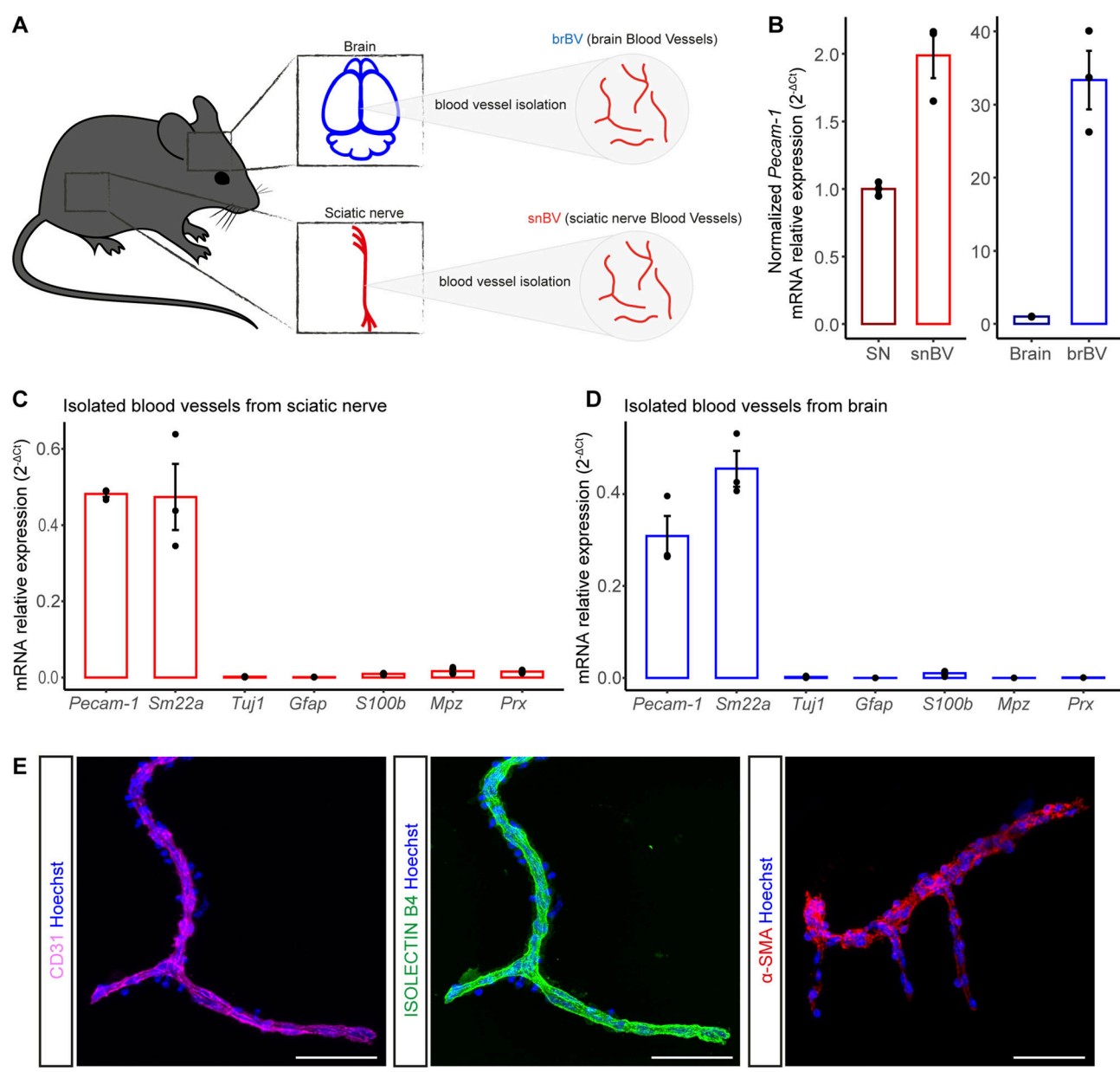

**Figure 1. Purified blood vessels from mouse brain and sciatic nerve.**
**(A)** Schematic representation of the distinct samples: brBV (for brain blood vessels) and snBV (for sciatic nerve blood vessels). **(B)** qRT-PCR of *Pecam-1* from isolated blood vessels and from entire brains or sciatic nerves (SN). **(C, D)** qRT-PCR of transcripts encoding CD31/PECAM1, SM22a, TUBB3, GFAP, S100 β, MPZ, and PRX from sciatic nerve (C) or brain (D) isolated blood vessels. **(E)** Immunofluorescence staining of purified snBV: endothelial cells are labeled in magenta (CD31) and in green (IB4), whereas mural cells including vascular smooth muscle cells are stained in red. All cell nuclei are labeled in blue (Hoechst). Data information: n = 3 samples. Each sample represents two mice (four sciatic nerves or two brains). Scale bar is 50 μm.

to *Pecam-1*/CD31 (Fig 2E, upper panels). Finally, a triple co-localization analysis enabled us to show that FABP4 was exclusively expressed along the veins of the sciatic nerve (sn), whereas arteries failed to show any staining (Fig 2E, lower panels). Furthermore, the strong staining of FABP4 along the veins failed to reveal any significant co-localization with either endothelial cells (*Pecam-1*/CD31) or vSMCs (α-SMA). Overall, these findings identify specific snBV markers that selectively label the distinct compartments or blood vessel types composing the *vasa nervorum*.

## Different molecular pathways characterise the adult sciatic nerve or brain blood vessels

To further characterise the comprehensive molecular signature of snBV and brBV, we performed gene ontology (GO) enrichment analysis with the objective of identifying differentially enriched pathways. Remarkably, three pathways related to "cilia," "central nervous system (CNS) functions," and "channels/transporters" turned out to be selectively significantly enriched in brBV (Fig 3A). If

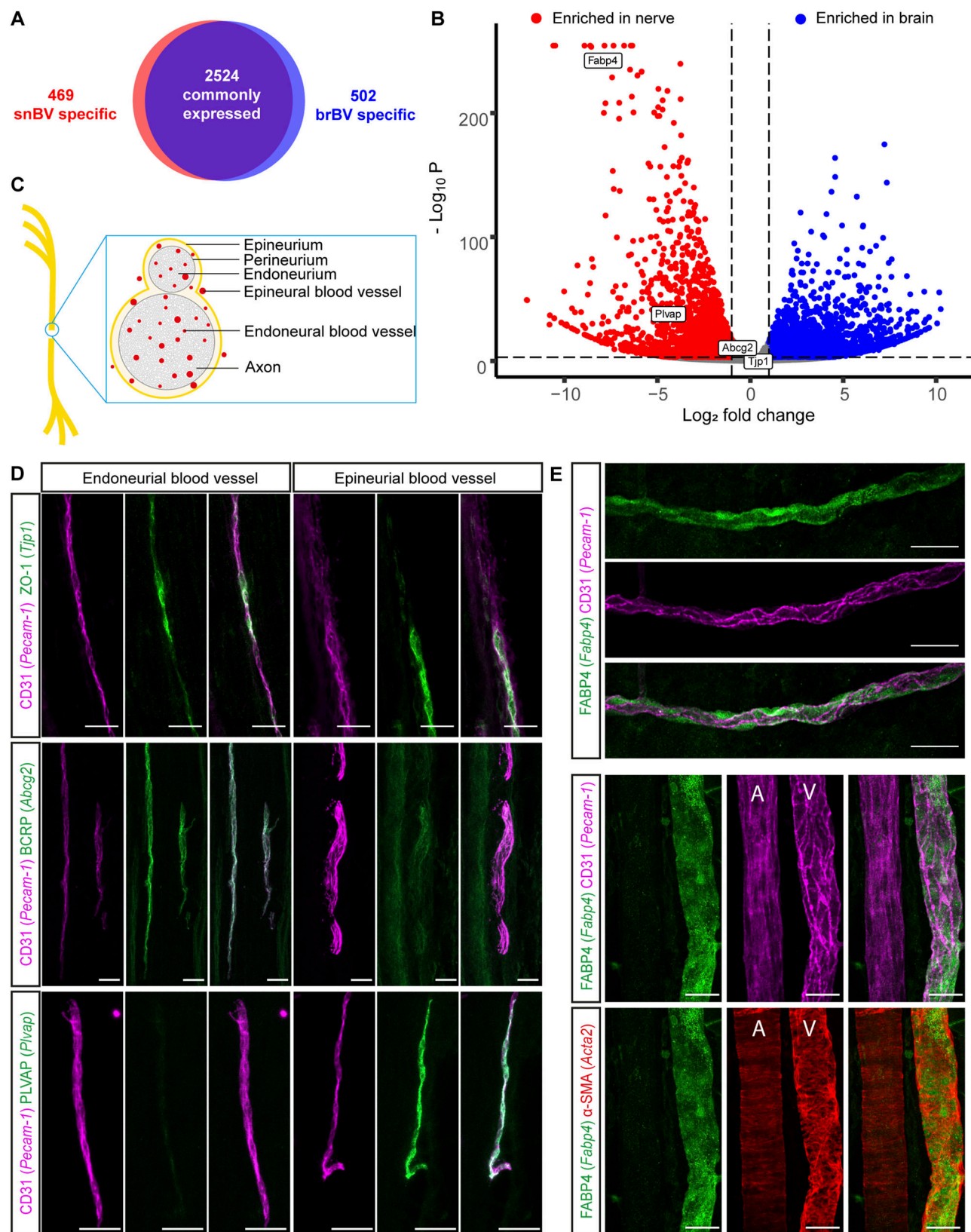

**Figure 2. Protein expression pattern of the *vasa nervorum*.**
**(A)** Number of commonly expressed, snBV (sciatic nerve blood vessels)- and brBV (brain blood vessels)-specific genes. Adjusted *P*-value cut-off <0.001; log$_2$ fold change < −1.5 or log$_2$ fold change > 1.5. Specific genes were considered when, in addition to those parameters, a given gene was not detected above 500 reads in the brain (snBV

the last two pathways were expected given the clear involvement of the BBB in various CNS functions and its large repertoire of transporters required for the fine-tuned crossing of molecules, a brain-specific "cilia" pathway turned out to be relevant with regard to recent studies revealing the presence of primary cilia in both endothelial and smooth muscle cells of brBV (Zhang et al, 2019; Thirugnanam et al, 2022). In terms of pathways significantly enriched in snBV, several of them, including those associated with "angiogenesis," "development/morphogenesis," "extracellular matrix," "signalling pathways," or "transcription/translation regulation" (Fig 3A) tend to characterise adult highly plastic vessels. These data may indeed indicate that snBVs are more prone than brBV to adapt, remodel and respond to homeostatic changes or pathogenic insult in view of their large range of enriched "developmental" genes related to cell plasticity. Furthermore, the heatmap (Fig 3B) showing the DEGs assembled by GO pathways further confirmed significant differences between the snBV and brBV vasculatures. In each pathway, some genes are commonly expressed at similar levels by both vasculatures, as shown for *Cldn5*/CLAUDIN-5, for example, in the "tight-junctions" dataset, whereas others (e.g., *Cldn1*/CLAUDIN 1) display highly differential expression patterns, being exclusively enriched in snBV or brBV (Fig 3B). Interestingly, some known oxaliplatin transporters (Fujita et al, 2019) appeared differentially expressed in the two vasculatures (Fig 3B): Octn2 (*Slc22a5*) and Oct3 (*Slc22a3*) were significantly more expressed in snBV than in brBV that reciprocally showed enriched Ctr1 (*Slc7a1*) or Mate1 (*Slc47a1*). These differences in oxaliplatin-transporter expression could partly account for the different drug permeability of the two barriers and the selective alteration of the peripheral nervous system in OIPN.

### The integrity of the vasa nervorum is preserved after oxaliplatin IV injections

To explore the impact of an intravenous (IV) oxaliplatin administration on the *vasa nervorum*, we induced neuropathic symptoms in mice by injecting three times oxaliplatin to reach a 30 mg/kg cumulative dose and carried out behavioral tests to assess their potential tactile allodynia and hypersensitivity to cold (Fig 4A). Remarkably, oxaliplatin-treated mice showed a decrease in average withdrawal threshold compared with control mice, which indicated the development of tactile allodynia (Fig 4B). Furthermore, the oxaliplatin-treated mouse group showed an increased number of paw withdrawal in response to exposure to a cold plate (4°C) compared with the control group, which revealed hypersensitivity to cold (Fig 4C). We thus established that oxaliplatin-treated mice developed significative neuropathic symptoms after three oxaliplatin IV injections with a cumulative dose of 30 mg/kg. We further showed that these neuropathic symptoms were reversible as

oxaliplatin-treated mice lost their tactile allodynia (Fig S1A) and cold hypersensitivity (Fig S1B) around 30 d after the first oxaliplatin injection. We also established that the symptoms observed with this cumulative dose of 30 mg/kg were conserved at 15 mg/kg (i.e., 3 injections of oxaliplatin at 5 mg/kg) but not at 6 mg/kg (3 injections of oxaliplatin at 2 mg/kg; Fig S2).

Because oxaliplatin was intravenously injected for patient use, we next investigated whether the overall morphology and composition of the mouse *vasa nervorum* was affected by an oxaliplatin IV administration. Three-dimensional (3D) reconstructions of the mouse sciatic nerve blood vessel network, based on CD31 immunostaining (Fig 4D), failed to reveal any major changes of the *vasa nervorum* between oxaliplatin-treated and control mice. Moreover, the subsequent quantifications of blood vessel length (Fig 4E) and branch point number (Fig 4F) per nerve did not show any significant differences between oxaliplatin-treated and control mice. These findings suggested that there was no vascular degeneration after oxaliplatin injections in our mouse model. To further explore the integrity of the *vasa nervorum*, we assessed the expression of both the endothelial cell marker CD31 (Fig 4G) and a major blood-nerve-barrier tight-junction protein (Ouyang et al, 2019), CLAUDIN-5 (Fig 4H). Both immunolabeling of CD31 and CLAUDIN-5 appeared continuous on the *vasa nervorum* of oxaliplatin-treated mice (Fig 4G and H). In addition, qRT-PCR of *Cldn5* mRNA relative expression from mouse sciatic nerve failed to reveal any significant differences between oxaliplatin-treated and non-treated mice (Fig 4I). Furthermore, a 3D reconstruction of CLAUDIN-5-labeled *vasa nervorum* (Fig S3A) did not show any statistical modifications in terms of blood vessel length (Fig S3B) and branch point number (Fig S3C) as shown on the CD31-stained BV network (Fig 4D).

As chronic administration of oxaliplatin not only causes OIPN in rats but may also give rise to a loss of nerve fiber endings in their epidermis (Boyette-Davis et al, 2011), we aimed to characterise the density of intraepidermal nerve fiber (IENF) in our mouse model. The immunostaining of mouse plantar hind paw skin with TUBB3 antibody, which turned here to allow visualization of more IENF compared with PGP9.5 immunostaining (Fig S4A), showed that there were no significant differences in IENF number, length, and volume between oxaliplatin-treated and non-treated mice (Fig S4B–E). Altogether, our data establish that both the *vasa nervorum* of the sciatic nerve and the IENF density remain intact post oxaliplatin injection during the acute OIPN phase, in which neuropathic symptoms are still reversible.

### The molecular landscape of the vasa nervorum is impacted by oxaliplatin

Using the purification protocol described in Fig 1A, we isolated the snBV from oxaliplatin-treated and sham-treated mice and carried

---

specific) or in the nerve (brBV specific). **(B)** Volcano plot of differentially expressed genes. Differentially expressed genes of interest are highlighted in white boxes. Adjusted *P*-value cut-off <0.001; $\log_2$ fold change < −1 or $\log_2$ fold change > 1. **(C)** Schematic representation showing the sciatic nerve structure. **(D)** Immunostaining on sciatic nerve longitudinal sections (14-$\mu$m thick). Blood vessels are stained in magenta (CD31/PECAM1). Two mutually expressed proteins, ZO-1 and BCRP and a nerve-enriched protein, PLVAP, are labeled in green. **(E)** Whole-mount immunostaining of a sciatic nerve. Endothelial cells are stained in magenta (CD31/PECAM1), smooth muscle cells in red ($\alpha$-SMA), and the differentially expressed and nerve-specific protein FABP4 in green. A and V are respectively for artery and vein. Data information: Scale bar 30 $\mu$m.

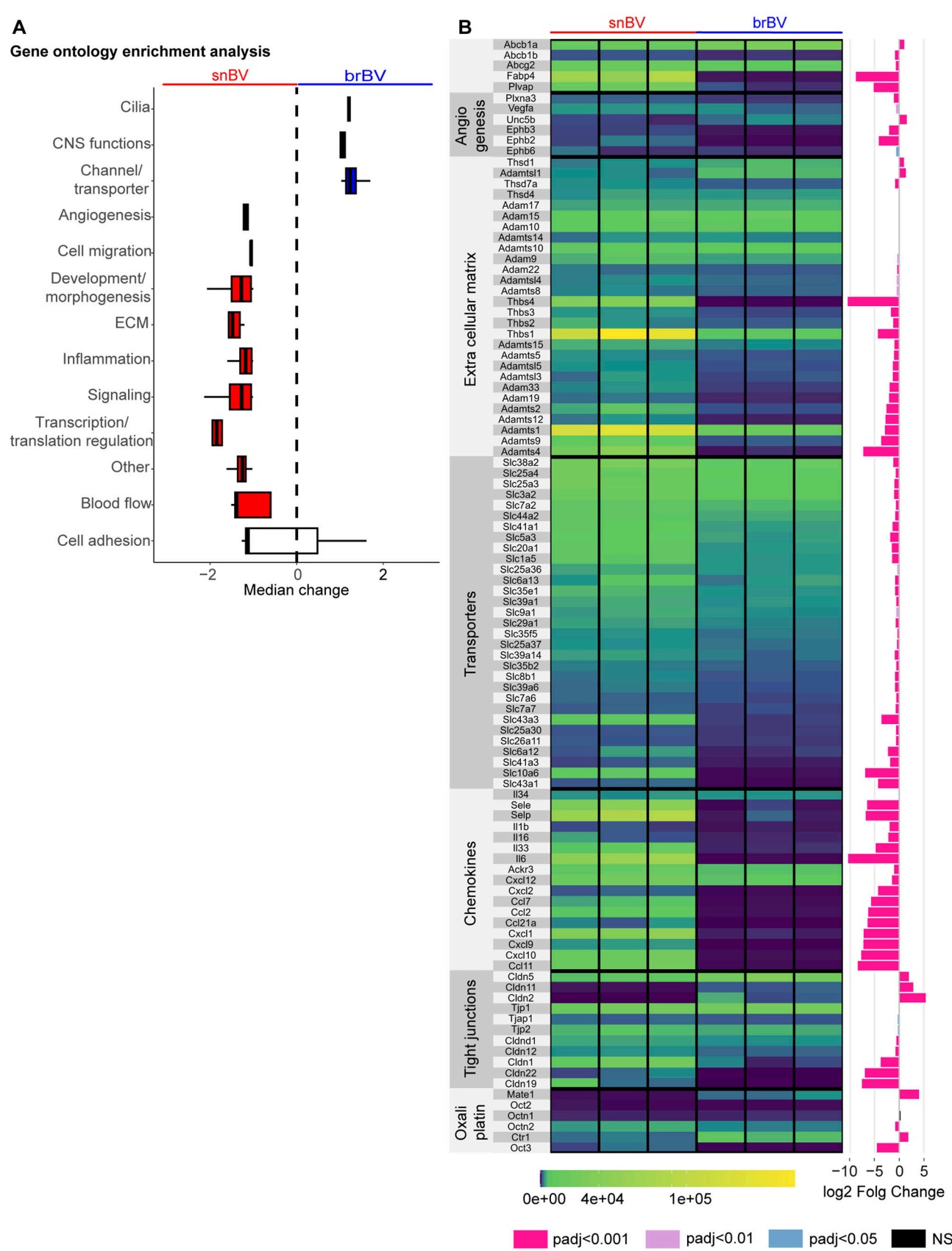

A

**Gene ontology enrichment analysis**

out a comparative transcriptomic analysis through bulk RNA se-quencing. In this comparison, samples were more homogenous as they were extracted from the nerve, and only differed from what IV was injected to the mice. In addition, as for the previous analysis, samples were made of heterogenous cell populations (EC and mural cells) which are known to underestimate differences be-tween genes expression. Finally, we thought that for a better un-derstanding of the etiology, as we were at the onset of OIPN symptoms, small molecular changes could still be meaningful. Thus, for this transcriptomic analysis, we modified our parameters. Our analysis resulted in the identification of 178 DEGs, including 53 down-regulated genes (adjusted *P*-value < 0.05, Log$_2$ Fold Change <0.5) and 125 up-regulated genes (adjusted *P*-value < 0.05, Log$_2$ Fold Change >0.5) in oxaliplatin-treated mice compared with control mice (Fig 5A). Notably, among the up-regulated DEGs, the corticosterone biosynthetic gene *Hsd11b1* (Fig 5A) was also found up-regulated in the bone marrow of rats treated with a combination of topotecan and oxaliplatin (Davis et al, 2015). Furthermore, the up-regulation of the microtubule-associated tumour suppressor gene *Mtus2*, shown to limit the metastatic activity of cancer cells (Kuo et al, 2017) and to express a long non-coding RNA *Mtus2-5* playing a key role in the angiogenesis of patients with a multifactorial thrombosis condition (Zhang et al, 2023), might be of interest as a downstream target of oxaliplatin IV treatment. Likewise, the up-regulated *Rasl10b* gene encoding a small GTPase with tumour suppressor potential (Zou et al, 2006), which was shown to regulate blood flow in the atrium (Rybkin et al, 2007), could also participate in the antitumoral response downstream of oxaliplatin. Thus, a number of identified DEGs seem promising for further investigation of oxaliplatin-induced molecular players and mechanisms un-derlying both the treatment efficiency and its detrimental conse-quences on the peripheral nervous system.

To characterise the biological pathways overrepresented after oxaliplatin administration, we performed a DEG enrichment anal-ysis of GO and KEGG terms using the Metascape portal (Zhou et al, 2019). Interestingly, four of the pathways identified in the enrich-ment analysis were related to blood flow control, such as "circu-latory system process," "oxytocin signalling pathway," "apelin signalling," and "vascular smooth muscle contraction" (Fig 5B). This analysis also pointed out that oxaliplatin seemed to specifically interfere with mural cell contractility, ion transport, intracellular adhesion, and extracellular matrix (Fig 5B). Altogether, this prompted us to hypothesize that oxaliplatin IV injections could alter the intra-nervous vasculature and thereby disrupt nerve homeostasis. To test this hypothesis, we further explored whether oxaliplatin treatment could selectively modify the expression pattern of its transporters, tight-junction proteins, and cues in-volved in vSMC contractile function as schematically summarized in Fig 5C. No significant differences in the expression levels of oxa-liplatin transporter genes (Fig 5D) nor in the major tight-junction

proteins of peripheral nerve endothelial cells (Fig 5E) were detected between oxaliplatin-treated and control animals. Thus, the transport of oxaliplatin did not tend to be increased at the onset of the pathology, neither its influx from the blood to the endothelial cells nor its efflux towards the blood or the parenchyma. Fur-thermore, the permeability of the intraneural vessels did not ap-pear to be altered either. Yet, several genes involved in the vSMC contraction pathway or more generally in vasoconstriction are significantly up-regulated in oxaliplatin-treated mouse *vasa nervorum* compared with that of control mice (Fig 5F), which was further confirmed for a subset of them by qRT-PCR on independent samples (Fig S5). Altogether, these results suggested that oxaliplatin interfered with snBV contractile function and could participate in OIPN development by restricting proper nerve perfusion.

## Vasodilators significantly alleviate oxaliplatin-induced neuropathic symptoms

To test the RNA-Seq prediction that oxaliplatin IV injection in-creased vasoconstriction of the *vasa nervorum*, which thereby led to the OIPN symptoms, we explored whether injecting two different vasodilators, tadalafil and ambrisentan, could counteract this deleterious effect of oxaliplatin treatment. Cues increasing vaso-dilation were previously reported to reduce some OIPN symptoms in mouse models (Gauchan et al, 2009; Ogihara et al, 2019). We thus chose to assess whether inhibiting the contraction using ambri-sentan, inducing vasodilation with tadalafil or a potential syner-gistic effect of both drugs could alleviate the peripheral symptoms of our OIPN mouse model (Fig 6A). We intraperitoneally injected 10 mg/kg tadalafil every other day whereas 5 mg/kg ambrisentan was injected every day as shown on the timeline (Fig 6B). We next performed behavioral tests in between each oxaliplatin injection to monitor the development and progression of neuropathic symp-toms (Fig 6B). Oxaliplatin-treated mice exhibited tactile allodynia after the first injection and until the end of the protocol (Fig 6C). The injection of ambrisentan significantly reduced oxaliplatin-induced tactile allodynia from the first oxaliplatin injection (Fig 6C). Likewise, tadalafil treatment, alone or in combination with ambrisentan significantly alleviated oxaliplatin-induced tactile allodynia, al-though from the second oxaliplatin injection onwards (Fig 6C). After the third IV injection of oxaliplatin and thus the end of the protocol, tadalafil seemed to more efficiently prevent oxaliplatin-induced allodynia than ambrisentan, alone or together with ambrisentan (Fig 6C). Furthermore, oxaliplatin-treated mice showed clear dis-comfort at lower temperatures compared with control mice (Fig 6D). When treated with ambrisentan or tadalafil, or with a combination of the two drugs, the oxaliplatin-induced cold hypersensitivity was significantly diminished after three oxaliplatin injections. Notably, the synergistic effect of the dual treatment with both ambrisentan and tadalafil seemed to more efficiently reduce the oxaliplatin-

---

**Figure 3. Molecular signatures of sciatic nerve and brain blood vessels.**
**(A)** Enriched pathways in snBV (sciatic nerve blood vessels) and brBV (brain blood vessels) gene sets. Mann–Whitney test, *P*-value <0.01. **(B)** Heatmap representation of differentially expressed genes of interest involved in different pathways: oxaliplatin transport, tight junctions, chemokines, transporters, extracellular matrix and angiogenesis. The five first differentially expressed genes on top are not associated with any specific pathways but were selected for their enrichment in snBV (*Fabp4*, *Plvap*) or commonly expressed in both tissues (*Abcg2*, *Abcb1a*, *Abcb1b*). Right: log$_2$ fold change between snBV and brBV. The color of the column refers to the adjusted *P*-value. Data information: n = 3 with 2 mice per sample.

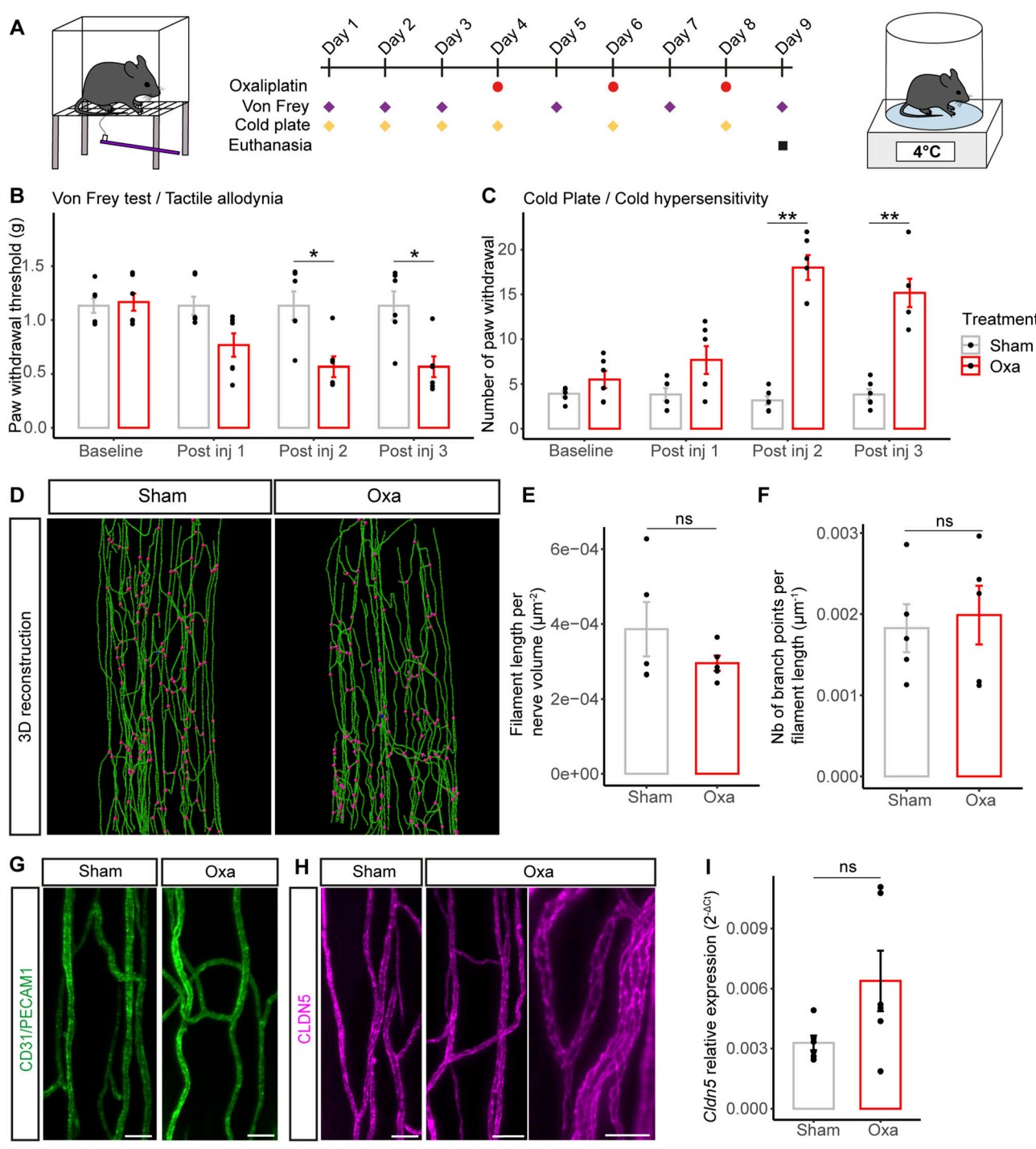

**Figure 4. Analysis of the *vasa nervorum* in mice developing neuropathic symptoms after oxaliplatin IV administration.**
**(A)** Schematic representation of oxaliplatin-induced peripheral neuropathy acute mice model representing oxaliplatin injection and behavioral test (Von Frey and cold plate) timepoints. **(B)** Paw withdrawal threshold before and after each injection of oxaliplatin (oxa) or glucose (sham) measured with Von Frey test. **(C)** Sensitivity to cold before and after each injection, with a temperature of 4°C during 2 min **(D)** 3D reconstruction of the blood vessel network based on CD31/Pecam-1 signal from cleared sciatic nerves. Branch points are represented in magenta. **(E)** Quantification of the blood vessel length per nerve volume. **(F)** Quantification of the number of blood vessel branch points per filament length. **(G)** 3D view (200-µm thick) of a sciatic nerve region from sham and oxa mice. CD31/PECAM1 is stained in green. **(H)** 3D view (200-µm thick) of a sciatic nerve region from sham and oxa mice (zoom on the right). CLDN5 is stained in magenta. **(I)** qRT-PCR of *Cldn5* mRNA. Data information; n = 5–6. Mann–Whitney test, two-tailed, *P < 0.05, **P < 0.01. **(B, C)** Supplemental test (not shown in graphs): Kruskal–Wallis test, (B): Sham ns, Oxa **; (C): Sham ns, Oxa ***, with *P < 0.05, **P < 0.01, ***P < 0.001. **(D, G, H)** Scale bar: (D): 100 µm, (G): 30 µm, (H): Left: 40 µm, zoom on the right panel: 20 µm.

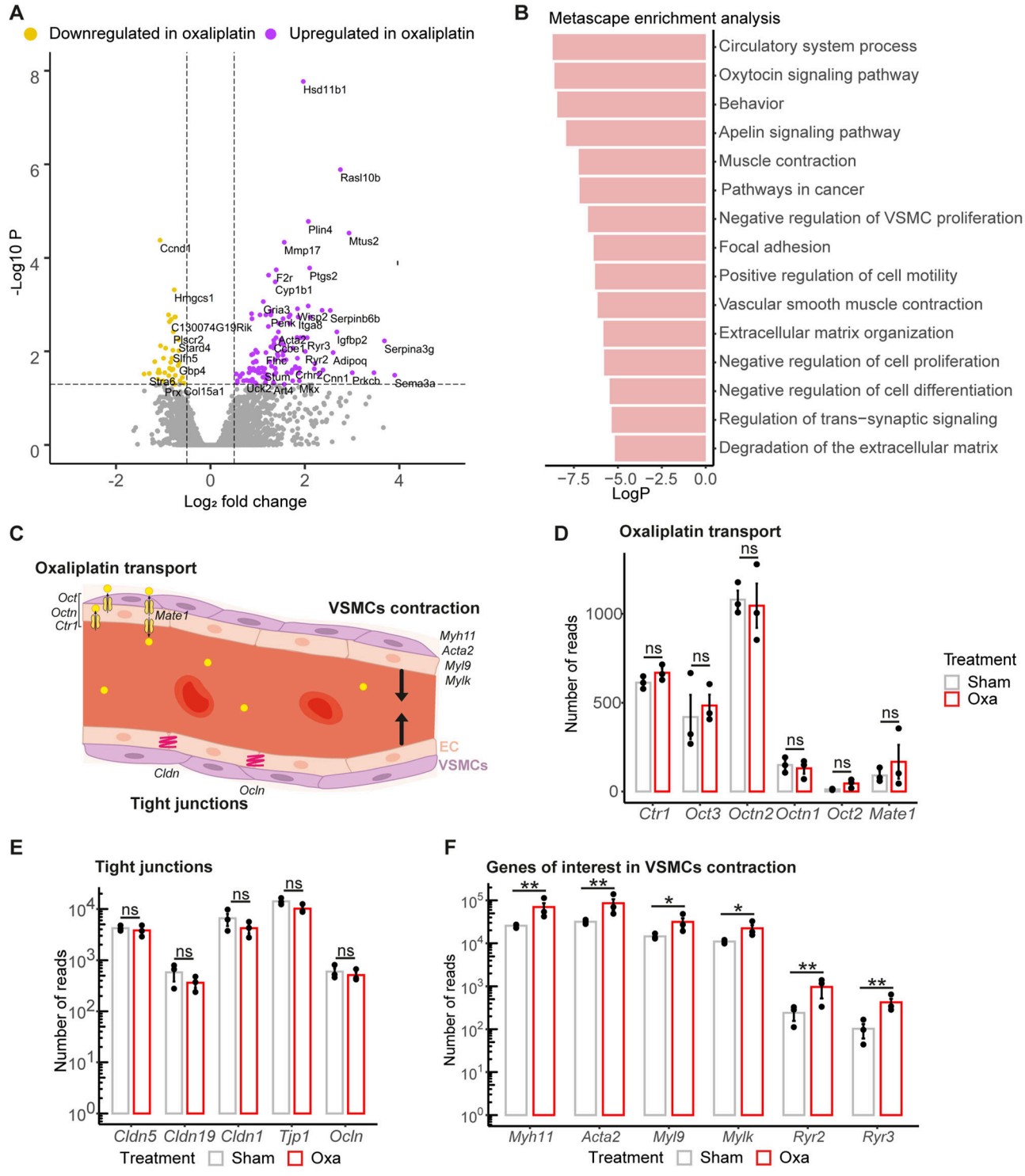

**Figure 5. Transcriptomic analysis of the *vasa nervorum* molecular composition in an acute mouse model of oxaliplatin-induced peripheral neuropathy.**
**(A)** Volcano plot representation of differentially expressed genes, adjusted *P*-value cut-off <0.05; log$_2$ fold change < −0.5 or log$_2$ fold change > 0.5. **(B)** Metascape analysis. Top 15 pathways enriched in differentially expressed genes. Low logP values correspond to the most enriched pathways. **(C)** Schematic representation showing genes of interest in blood vessel biological processes. **(D, E, F)** Number of reads for mRNA encoding oxaliplatin transporters/tight-junction proteins/genes of interest involved in vascular smooth muscle contraction from RNA sequencing data. Data information: n = 3 with 2 mice per sample, adjusted *P*-value from DESeq2 analysis *P < 0.05, **P < 0.01.

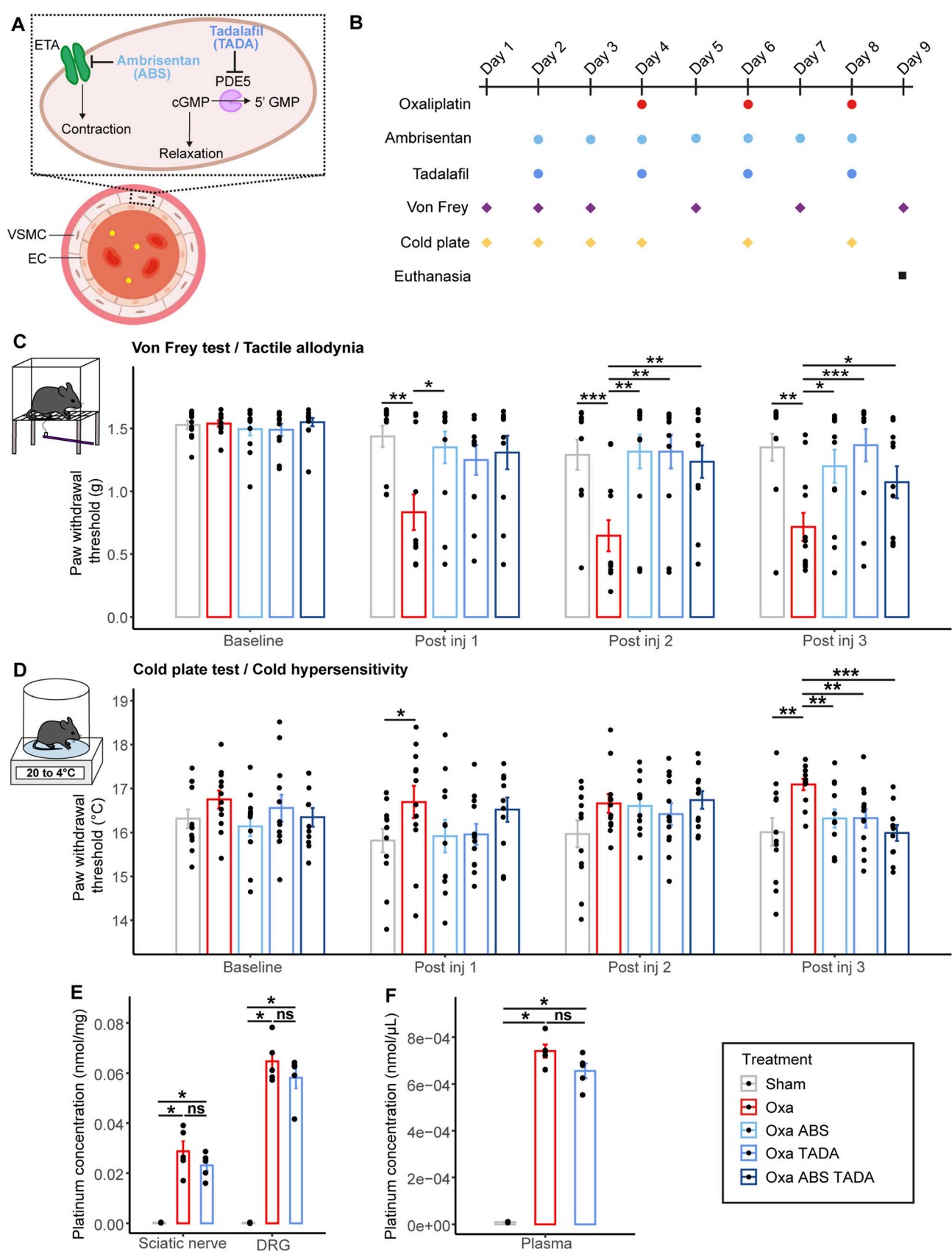

induced cold hypersensitivity compared with the single use of tadalafil or ambrisentan (Fig 6D). We next ensured that the administration of one of these vasodilative molecules, tadalafil, did not interfere with the availability of oxaliplatin in various tissues, for example, by blocking the transporters involved in the oxaliplatin influx or efflux. We did not observe any significant differences in platinum concentrations within the sciatic nerves, the dorsal root ganglias (DRGs) (Fig 6E) and the plasmas (Fig 6F) of oxaliplatin-treated mice and doubly treated (oxaliplatin + tadalafil) mice, suggesting that tadalafil administration did not interfere with oxaliplatin biodistribution. These results thereby suggested that both ambrisentan and tadalafil could significantly alleviate oxaliplatin-induced neuropathic symptoms, although with different efficacy depending on the symptoms analyzed. Because the bioavailability of oxaliplatin was not altered by the administration of tadalafil, this strongly suggests that vasodilators reduce oxaliplatin-induced neuropathic symptoms by increasing nerve perfusion and not by modifying oxaliplatin accumulation in tissues.

### Ambrisentan and tadalafil reduce oxaliplatin-induced nervous hypoxia

The observation that vasodilators significantly reduced oxaliplatin-induced tactile allodynia and cold hypersensitivity in mice suggested that oxaliplatin could perturb nerve homeostasis via reduced perfusion. A direct consequence of a reduced blood perfusion rate in the nerve could be that the nervous tissue becomes hypoxic. To test this hypothesis, we carried out sciatic nerve immunostaining with the transcription factor HIF-1α (Fig 7A), a key marker for hypoxia (Guillemin & Krasnow, 1997). Because the reduction in peripheral nerve area may reveal nerve damage (Koike et al, 2010), we estimated the HIF-1α-fluorescent-intensity-to-sciatic-nerve-area ratio in oxaliplatin-treated mice with or without vasodilative treatment (Fig 7B). Remarkably, oxaliplatin-treated mouse sciatic nerves showed a significant increase in HIF-1α levels (Fig 7A and B). Moreover, the co-administration of oxaliplatin with vasodilative molecules (tadalafil and/or ambrisentan) drastically reduced oxaliplatin-induced hypoxia of the sciatic nerve (Fig 7A and B). It should here be noted that because tadalafil and ambrisentan individually rescued oxaliplatin-induced increase in HIF-1α fluorescent intensity, no synergistic effect could be detected for the dual injection of these vasodilators (Fig 7B). These results were next confirmed using Western blot analysis (Fig 7C). Indeed, a lower dose of oxaliplatin (3 injections of 5 mg/kg), which was sufficient to induce neuropathic symptoms (Fig S2), caused nerve hypoxia reversed by tadalafil administration (Fig 7C). To further assess the involvement of the HIF-1α pathway, we analyzed the expression of a HIF-1α downstream effector (Liu et al, 1995; Forsythe et al, 1996),

VEGF-A. Using qRT-PCR and Western blot analyses, we observed a significant increase in the transcription of *Vegfa* (Fig 7D) and an increase in VEGF-A protein expression (Fig 7E) in oxaliplatin-treated mouse sciatic nerves. Notably, an increase in the VEGFR-1 receptor was also detected by co-immunohistochemistry in intra-nervous blood vessels of oxaliplatin-treated sciatic nerves. The recruitment of this receptor to the nerve blood vessels is importantly reversed by tadalafil administration (Fig S6A and B).

## Discussion

To explore the molecular specificity of the BNB compared with one of the most selective barrier of the body, the BBB, we developed a protocol to purify intact fragments of the *vasa nervorum* from adult mouse sciatic nerves and conducted a comparative transcriptomic analysis between brain and sciatic nerve vasculature. Notably, among numerous DEGs shown to be specifically expressed, or significantly enriched, either in brain BVs (brBVs) or in sciatic nerve BVs (snBVs), our analysis revealed the selective expression of some transporters responsible for the uptake of a chemotherapeutic drug, oxaliplatin, in brBVs or snBVs. Indeed, as for other platinum salts used as anti-cancer drugs known to cause a common and dose-limiting peripheral neuropathy, several transporters localized to the cell membrane of dorsal root ganglion were associated with oxaliplatin uptake and efflux across the plasma membrane: the organic cation transporters OCT2, OCT3, and OCTN1/2, as well as the copper transporter CTR1 were described to be preferentially involved in oxaliplatin uptake, whereas copper P-type ATPase ATP7B as well as multidrug resistance-associated protein 2 (MRP2) and multidrug and toxin extrusion protein 1 (MATE1) would mediate the efflux of oxaliplatin (Fujita et al, 2019; Gu et al, 2019; Cheng et al, 2023). Given that oxaliplatin treatment exclusively affects the peripheral nervous system, we searched for any selective expression of these transporters in snBVs, which could underlie the pathogenesis of OIPN. Interestingly, whereas *Mate1* and *Ctr1* transcripts are significantly enriched in brBVs, *Octn2* and *Oct3* mRNAs are reciprocally up-regulated in snBVs. These different levels of oxaliplatin transporter mRNAs may participate in underpinning the selective toxicity of oxaliplatin onto peripheral nerves, particularly because the genetic or pharmacological targeting of OCT2 was recently shown to improve acute and chronic forms of neurotoxicity in a mouse model of OIPN (Huang et al, 2020). However, a therapeutic approach aiming at targeting OCT2 to simultaneously prevent OIPN (Nepal et al, 2022) and increase oxaliplatin efficiency in some cancer treatment (Liu et al, 2016) might be difficult to implement without further studies in both animal models and cohorts

---

**Figure 6. Preventive administration of ambrisentan and tadalafil improves oxaliplatin-induced neuropathic symptoms without affecting its accumulation.**
**(A)** Schematic representation of ambrisentan and tadalafil molecular targets. **(B)** Schematic representation of the timepoints in oxaliplatin, ambrisentan, tadalafil injections and behavioral tests (Von Frey and cold plate). **(C)** Paw withdrawal threshold before and after injections of oxaliplatin (oxa), ambrisentan (ABS), or tadalafil (TADA) or both (Oxa ABS TADA). **(D)** Sensitivity to cold before and after injections of oxaliplatin, ambrisentan, and tadalafil. The temperature starts à 20°C and the plate loses 2°C per min. The threshold represents the temperature of the first response to cold. **(E, F)**: Platinum concentrations in sciatic nerves, dorsal root ganglias (E) and plasma (F) from sham, oxaliplatin-treated or doubly oxaliplatin + tadalafil–treated animals. **(C, D, E, F)** Data information: (C, D) n = 12 (except n = 11 for Oxa ABS TADA at post inj 3), (E, F) n = 5 (except n = 3 for sham). Mann–Whitney test, two-tailed: *$P < 0.05$, **$P < 0.01$, ***$P < 0.001$. Supplemental tests (not shown in graphs). (C): Kruskal–Wallis Sham ns, Oxa ****, Oxa ABS ns, Oxa TADA ns, Oxa ABS TADA. *(D): Kruskal–Wallis Sham ns, Oxa ns, Oxa ABS ns, Oxa TADA ns, Oxa ABS TADA ns. *$P < 0.05$, **$P < 0.01$, ***$P < 0.001$.

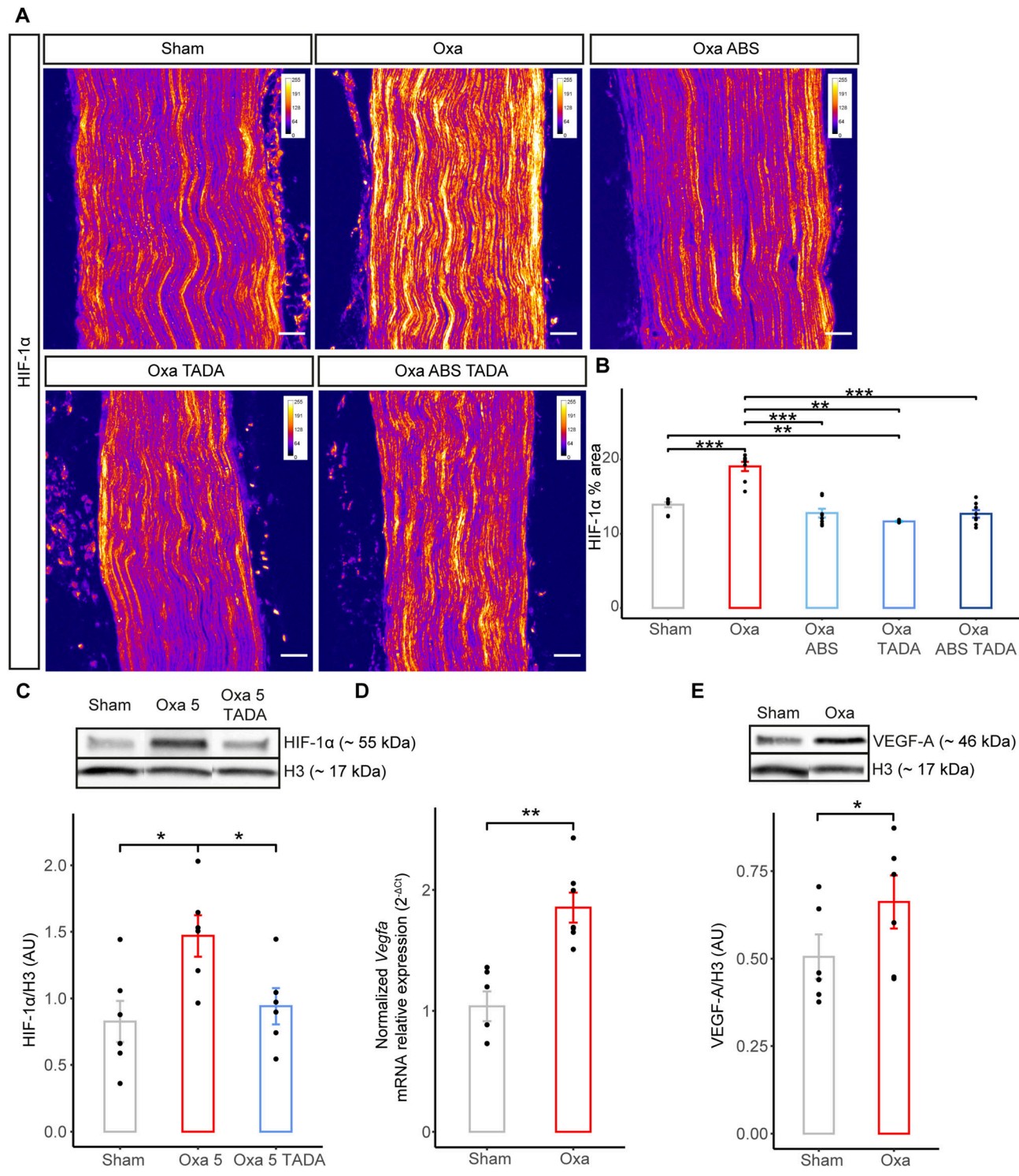

**Figure 7.  Administration of ambrisentan and tadalafil reduces oxaliplatin-induced nerve hypoxia and HIF-1α downstream effector VEGF-A.**
**(A)** Immunostaining of sciatic nerve longitudinal sections with anti-HIF-1α antibody. Nerves dissected from sham and oxaliplatin-treated animals (10 mg/kg) or treated with oxaliplatin (10 mg/kg) + tadalafil (Oxa TADA) or ambrisentan (Oxa ABS) or tadalafil + ambrisentan (Oxa ABS TADA) were analyzed. Representative images for each condition are visualized using an intensity-based color-coded representation using imageJ (fire LUT lookup table) to assess HIF-1α intensity. Colors represent signal intensity scale in the range of 0–256 (arbitrary unit) for each condition. **(B)** Quantification of HIF-1α-positive areas normalized to the nerve area. **(C)** Western blot analysis of HIF-1α expression in whole nerve homogenate from mice treated with oxaliplatin (5 mg/kg), oxaliplatin (5 mg/kg) + tadalafil (10 mg/kg) and glucose (sham). **(D)** qRT-PCR analysis of *Vegfa* expression from whole sciatic nerve of oxaliplatin-treated mice (10 mg/kg) compared with control mice (sham). **(E)** Western blot analysis of VEGF-A expression in whole nerve homogenate from mice treated with oxaliplatin (10 mg/kg) or glucose (sham). Data information: n = 4–8, Mann–Whitney test, two-tailed, *P < 0.05, **P < 0.01, ***P < 0.001. Scale bar 50 µm. **(E)** Mann–Whitney test, two-tailed, *P-adjusted < 0.05.

of patients (Winter et al, 2017; Zheng et al, 2017). These data prompted us to further characterise the anatomy and the molecular composition of the *vasa nervorum* in a mouse model of OIPN.

Immunostaining analyses of 3D-transparized sciatic nerves allowed us to show that the global architecture of snBVs was not affected by acute oxaliplatin administration, although the mice exhibited robust neuropathic symptoms. So far, most mouse models of acute OIPN have been developed by single or repeated intraperitoneal injection(s) of oxaliplatin (Gauchan et al, 2009; Descoeur et al, 2011; Ogihara et al, 2019; Gould et al, 2021; Wang et al, 2023). Aiming at mimicking the patients' oxaliplatin administration and assessing the direct effect of oxaliplatin onto EC and mural cells composing the *vasa nervorum*, we investigated the relevance of a vascular component as the source of OIPN symptom initiation via intravenous injections of oxaliplatin. Our oxaliplatin-injected mice displayed notable mechanical allodynia and cold hypersensitivity, which confirmed the behavioral analyses of previous mouse models injected intraperitoneally with oxaliplatin (Descoeur et al, 2011; Ogihara et al, 2019; Gould et al, 2021). Furthermore, the intraperitoneal administration of a vasodilator, tadalafil, or an inhibitor of vasoconstriction, ambrisentan, was sufficient to significantly prevent acute OIPN symptom occurrence as revealed by both Von Frey and cold plate behavioral tests. These findings imply that oxaliplatin induces reversible enhanced vasoconstriction as BVs are not physically altered but only impaired in their vasoreactivity, which may be rescued by the use of distinct vasodilating drugs (Gauchan et al, 2009; Ogihara et al, 2019; present study).

This early vascular dysfunction is corroborated by our comparative transcriptomic analysis of oxaliplatin- or vehicle-treated sciatic nerve BVs, which revealed a significant up-regulation of the vascular contraction pathway. Indeed, several genes involved in the regulation of vasoconstriction, including *Myh11*, *Acta2*, *Myl9*, and *Mylk* encoding structural proteins of vSMC contractile unit (Karimi & Milewicz, 2016) or *Ryr2* and *Ryr3* encoding the ryanodine receptors that participate in regulating vSMC contractility (Kaßmann et al, 2019; Pritchard et al, 2019) are all up-regulated in oxaliplatin-injected sciatic nerve BVs compared with controls. Interestingly, additional genes found up-regulated in oxaliplatin-administered mouse nerve BVs, such as *Grip2* and *Cnn1*, were respectively identified as regulators of vSMC contractility in cerebral arteries (Fouillade et al, 2013) and the aorta (Castro et al, 2012). These data are not only consistent with the rescue of oxaliplatin-induced neuropathic pain using vasodilators in our mouse model of acute OIPN but also suggest that this enhanced vasoconstriction may lead to a decrease in nerve perfusion and possibly hypoxia that would evidently perturb peripheral nerve homeostasis.

To assess the level of nerve hypoxia after administration of oxaliplatin, we explored the expression of the transcription factor HIF-1α (Guillemin & Krasnow, 1997) on intact sciatic nerves from oxaliplatin-treated mice with or without vasodilator administration. This analysis showed a substantial increase in HIF-1α, reflecting the nerve hypoxic state, in oxaliplatin-administered mice, whereas the vasodilating drugs restored nerve normoxia. An increase in HIF-1α subsequently triggered the increase in VEGF-A expression in our oxaliplatin-treated mice. Notably, an overexpression of HIF-1α was detected by Western blot analysis from oxaliplatin-injected mouse sciatic nerves (Yang et al, 2019) and dorsal root ganglia (Wang et al, 2023), whereas this microcirculation hypoxia was associated with the formation of neutrophil extracellular traps (Wang et al, 2023). Interestingly, the increased levels of VEGF-A have been associated with neuropathic pain, through the activation of the neuronal VEGFR-1 signalling (Micheli et al, 2021). The increase in VEGF-A levels in our OIPN model could as well sensitize these neuronal receptors to induce their downstream signalling. In addition, we observed the recruitment of the VEGFR-1 receptor to the blood vessels of oxaliplatin-treated animals. The activation of the VEGF-A/VEGFR-1 pathway may lead to increased permeability of intra-nervous blood vessels (Shibuya, 2011), which could be implicated in the transition from acute to chronic phases of the disease. Importantly, the hypoxic response triggers the up-regulation of several genes, including those encoding α- and β-globins (Grek et al, 2011), the 5-amino-levulinate synthase (ALAS2, Hofer et al, 2003), interleukin-1β (IL1β, Gui et al, 2016), and apelin (APLN, Zhang et al, 2016) in various model systems. Yet, in our acute OIPN mouse model, seven of these genes, *Hbb-bs*, *Hba-a2*, *Hbb-bt*, *Hba-a1*, *Alas2*, *Il1b*, and *Apln*, turned out to be strikingly down-regulated in oxaliplatin-treated mouse nerve compared with controls. This finding, which partly matches a recent study showing a significant down-regulation of *Hba-a1*, *Hba-a2*, and *Alas2* in the dorsal root ganglia of mice injected with a single dose of oxaliplatin (Gould et al, 2021), further implicates the down-regulation of these genes in neuropathic pain physiopathology and suggests that oxaliplatin may participate in blocking the hypoxic response in addition to its induction of nerve vasoconstriction. Remarkably, alpha-globins were shown to control hypoxia-induced vasodilation by generating nitric oxide (NO), at the origin of the most potent vasodilatory pathway, via nitrite reduction (Keller IV et al, 2022). One could thus speculate that the lack of alpha-globin nitrite-reductase activity in our OIPN mouse model prevents the physiological hypoxia/NO-induced vasodilation, which led to the occurrence of neuropathic symptoms that may be rescued by an additional intake of vasodilators.

In conclusion, our molecular and functional results in a mouse model mimicking the administration of oxaliplatin in patients reveal a new role for blood flow regulation, which are essential for preventing or alleviating the onset of acute symptoms of OIPN. Our comprehensive analysis of the *vasa nervorum* provides strong evidence for a role of vascular dysfunction, leading to nerve hypoxia, in the early pathogenesis of OIPN. Remarkably, microvascular dysfunction and endoneurial hypoxia were shown to contribute to the genesis of neuropathic pain associated with traumatic nerve injury (Lim et al, 2015) and diabetic peripheral neuropathy (Tuck et al, 1984; Østergaard et al, 2015). Recently, the stabilization or overactivation of the HIF-1α signalling pathway, reflecting peripheral nerve hypoxia, has been similarly linked to the pathogenesis of both bortezomib- or paclitaxel-induced peripheral neuropathy (Ludman & Melemedjian, 2019; Kober et al, 2020). Our work thus opens up new therapeutic avenues for the treatment of different forms of CIPN in emphasizing the vascular contribution to nerve homeostasis and in providing preclinical evidence for vasomotion treatment repurposing. Evaluating the efficacy of vasodilators in chronic models of OIPN might be

critical to assess the duration of their preventive effect and investigate their innocuity on the primary antitumoral role of oxaliplatin in cancer mouse models.

# Materials and Methods

### Animals

We used male C57BL/6JRj (Janvier-Labs) mice, aged 3–6 mo. All experiments and techniques complied with the ethical rules of the French Agency for Animal Experimentation (project numbers #11822, #30827, and #38199).

### Drug administration

Oxaliplatin (Accord, kindly provided by Dr Salvatore Cisternino) was diluted in 5% glucose solution for a final concentration of 2, 5, or 10 mg/kg. A total of 100 $\mu$l was intravenously injected three times (i.e., a respective cumulative dose of 6, 15, or 30 mg/kg) to the animals. Animals from the sham group received similar injections of 5% glucose. Tadalafil (SML1877; Sigma-Aldrich) was resuspended in DMSO at a concentration of 5 mg/ml and given every other day by IP injection of 10 mg/kg, starting 2 d before the first oxaliplatin injection. Ambrisentan (SML2104; Sigma-Aldrich) was equally resuspended in DMSO at a concentration of 2 mg/ml and given every day by IP injection of 5 mg/kg, starting 2 d before the first oxaliplatin injection.

### The Von Frey test

Mice were individually placed in plastic boxes on a wire mesh floor and habituated for 30 min before testing. Von Frey filaments (BioSeb) of different binding forces (from 0.07, 0.16, 0.4, 0.6, 1, to 1.4 g) were applied five times to the mid-plantar skin of each hind paw. A reaction was counted as positive if the mouse immediately lifted the paw in reaction to the filament. For a given filament strength, if the number of positive reactions was equal or higher than 3/5, the mechanical threshold was considered to be reached. If a mouse did not respond to the 1.4 g filament, an arbitrary score of 1.6 was assigned.

### Cold plate test

Mice were placed in a clear plastic box and first habituated to the cold plate apparatus (Bioseb) primarily set at RT (25°C), then at 4°C. For measurement, mice were submitted to the cold plate at 4°C during 2 min and the number of responses to cold hyperalgesia was counted (paw lift, licking, jump). At the request of the French Ethical Committee, we next modified the test according to the following protocol: mice were placed on the plate at an initial temperature of 20°C, and the plate was programmed to reduce by 2°C/min. The first discomfort reaction shown by the mouse was recorded as the cold tolerance threshold.

### Clearing and immunostaining of entire sciatic nerves

After euthanasia, sciatic nerves were dissected and fixed in 4% PFA overnight at 4°C. We next used the iDisco+ staining and clearing method (Renier et al, 2014). After three PBS washes, samples were permeabilized with a solution of PBS/0.2% Triton/2.3% Glycine/2% DMSO for 2 d at 37°C followed by a blocking step with a solution of PBS/0.2% Triton/10% DMSO/6% donkey serum for 2 d at 37°C. Nerves were then incubated with primary antibodies diluted in PBS/0.2% Tween 20/1% heparin (10 mg/ml)/5% DMSO/3% donkey serum during 3 d at 37°C. After five washes with PTwH solution (PBS/0.2% Tween 20/1% heparin), samples were incubated with secondary antibodies in PTwH/3% donkey serum during 3 d at 37°C. After 5 washes with PTwH, nerves were included in 1% agarose blocks. Dehydration was performed with methanol/$H_2O$ series (20%, 40%, 60%, 80%, and 100% for 1 h each) at RT and incubated in 66% dichloromethane (DCM, Sigma-Aldrich)/33% methanol during 3 h. This was followed by two incubations with 100% DCM for 15 min each. Samples were next cleared with dibenzyl ether (DBE, Sigma-Aldrich) overnight. The list of all the antibodies used in this study is summarized in Table 1.

### Quantification of cleared sciatic nerves vascularization

Cleared sciatic nerves were imaged with an ultramicroscope (Lavision Biotech), and the Imaris software (Bitplane) was used to develop a 3D reconstruction of the CD31 and CLDN5 staining with the function "surface" followed by "filament tracer" to quantify the total length and the branch point number of the vessel network.

### Whole-mount and cryosection immunofluorescent staining

After euthanasia, sciatic nerves were dissected and fixed in a 4% PFA solution during 30 min at RT. The whole mounts were incubated in TNBT solution composed of Tris–HCl pH7.4/5 M NaCl/0.5% blocking reagent (Perkin)/0.5% Triton X-100 overnight at 4°C. Primary antibodies were diluted in the same solution and left with the samples overnight at 4°C. After washes with TNT solution (Tris–HCl pH7.4/5 M NaCl/0.05% Triton X-100), nerves were incubated with secondary antibodies diluted in TNBT solution during 3 h at RT. After washes with TNT solution and a final wash in PBS, entire sciatic nerves were mounted with Fluoromont (Dako). For cryosections, immediately after being dissected, sciatic nerves were embedded in OCT media and snap-frozen in liquid nitrogen, except for HIF-1$\alpha$ staining in which sciatic nerves were fixed in PFA overnight at 4°C and cryopreserved in 15% and 30% sucrose baths. Cryostat sections (14 $\mu$m) were fixed using ice-cold 100% methanol for 8 min then incubated in a blocking solution composed of 0.25% Triton X-100/10% FBS/PBS over 30 min at RT. Primary antibodies were diluted in the same solution, in which sections were incubated overnight at 4°C. After PBS washes, secondary antibodies were diluted in a 0.1% Triton X-100/1% FBS/PBS solution where sections were incubated for 2 h at RT. For VEGFR-1 staining, sciatic nerves were dissected and fixed in a 4% PFA solution during 4 h then cryopreserved in a 30% sucrose bath at 4°C overnight. Cryostat sections (14 $\mu$m) were permeabilized in 0.3% Triton X-100/PBS during 30 min at RT. After washes in PBS, sections were incubated in a blocking solution (10%

| Designation | Supplier and reference | Dilution | Structure/function |
|---|---|---|---|
| Anti-CD31/PECAM1 | BD Biosciences/Pharmingen | 1:200 (blood vessel staining) | Endothelial cells |
| | | 1:400 (Whole mount, sciatic nerve cryosections) | |
| Anti-α-SMA Cy3 | Sigma-Aldrich | 1:200 (blood vessel staining) | Smooth muscle cells |
| | | 1:400 (entire sciatic nerve) | |
| Anti-isolectine B4 (IB4) 488 | I21411; Invitrogen | 1:100 | Vascular membrane |
| Anti-PLVAP | 553849; BD Pharmingen | 1:200 | Vascular transport |
| Anti-FABP4 | 15872-1-AP; Proteintech | 1:200 | Vascular transport |
| Anti-ZO-1 | 20742-1-AP; Proteintech | 1:200 | Tight junction in EC |
| Anti-BCRP | ALX-801-036; Enzo Life Sciences | 1:200 | Vascular transport |
| Anti-CLAUDIN-5 | 34–1600; Thermo Fisher Scientific | 1:500 | Tight junction in EC |
| Anti-HIF-1α | NB100-449SS; Novus | 1:200 (IF) | Hypoxia-inducible factor 1 alpha |
| | | 1:500 (WB) | |
| Anti-PGP9.5 | 15503; Abcam | 1:400 | Neuronal biomarker |
| Anti-TUBB3 | 201202; BioLegend | 1:400 | Neuron cytoskeletal |
| Anti-VEGFR-1 | AF471; R&D | 1:400 | Vascular endothelial growth factor receptor 1 |
| Anti-VEGF-A | AB46154; Abcam | 1:500 | Vascular endothelial growth factor A |
| Anti-H3 | 14269; Ozyme | 1:1,000 | Histone H3 |
| Alexa fluor goat anti-rat 555 | A21434; Invitrogen | 1:400 | |
| Alexa fluor donkey anti-rabbit 488 | A21206; Invitrogen | 1:400 | |
| Alexa fluor donkey anti-rabbit 647 | A31573; Invitrogen | 1:500 | |
| Alexa fluor donkey anti-rabbit 555 | A31572; Invitrogen | 1:400 | |
| Alexa fluor goat anti-mouse IgG2a 555 | A21137; Invitrogen | 1:400 | |
| Alexa fluor donkey anti-goat 555 | A21432; Invitrogen | 1:400 | |
| Anti-rabbit IgG HRP | HAF008; R&D | 1:5,000 | |
| Anti-mouse IgG HRP | G-21040; Invitrogen | 1:5,000 | |

donkey serum/PBS) during 1 h at RT. Primary antibodies were diluted in the same solution, in which sections were incubated overnight at 4°C. After PBS washes, secondary antibodies were diluted in a 1% BSA/0.1% Triton X-100/PBS solution where sections were incubated for 1 h at RT.

### Immunostaining of IENF

Mice were euthanized by cervical dislocation and the paw skins were dissected and fixed in 4% PFA overnight at 4°C. These skins were then rinsed twice in PBS before being submitted to successive incubations at 10%, 20%, and 30% sucrose/PBS at 4°C for cryopreservation. The paw skins were next included in OCT and frozen in liquid nitrogen before being sectioned using a cryostat (25 or

100 $\mu$m). The sections were thawed for 30 min at RT, blocked for 30 min in 10% BSA/PBS and incubated overnight à 4°C with the primary antibody solution diluted in 1% BSA solution/0.3% Triton X-100. The slices were next rinsed three times in PBS and incubated for 2 h at RT with secondary antibodies diluted in 1% BSA. After two PBS baths, the sections were incubated with Hoechst (1:1,500) for 5 min and mounted with Fluoromont (Dako) after three baths of PBS.

### IENF quantification

Plantar paw skins (three to five sections per animal, n = 5) were imaged with Zeiss confocal LSM980 (40X lens). Using Fiji, the epidermal area was defined with the "Freehand Line" tool as the

nucleus-dense area and we counted the number of terminal nerve fibers labeled with the βIII tubulin (TUBB3) antibody that crossed this area. The density of IENF was evaluated by making the ratio of the number of TUBB3-positive fibers over the area of the epidermis. The length and volume of the fibers were quantified using Imaris software (Bitplane) and a 3D reconstruction of the TUBB3 staining within the epidermis area was developed with the function "surface."

## Sciatic nerve dissociation

After dissection, sciatic nerves were cut into 5–10 small pieces using scissors and incubated in a digestion solution (DMEM/FBS 10% with 5 mg/ml Collagenase type II, C2-22; Sigma-Aldrich) heated to 37°C for 25 min in a thermocycler with gentle agitation. Before starting, small pieces were mechanically dissociated by flushing them up and down 10 times with a 21 g needle. Then, every 5 min, mechanical dissociation was performed by gentle flushing up and down 10 times using a 26 G needle. After digestion, cells were pelleted by centrifugation at 15,300$g$ and at 4°C for 10 s and resuspended in TRIzol reagent (Invitrogen) for RNA extraction.

## Purification of brain (brBV) and sciatic nerve (snBV) blood vessels

A protocol used for the purification of brain blood vessels (Boulay et al, 2017) was here adapted to the *vasa nervorum*. Briefly, sciatic nerves (four sciatic nerves were pooled for one preparation to generate enough material for transcriptomic analysis), and brains were enzymatically digested with Liberase DL (Roche) at 143 $\mu$g.ml$^{-1}$ and DNAse I (Sigma-Aldrich) at 107 $\mu$g.ml$^{-1}$ at 37°C for 45 min. After a centrifugation for 10 min at 2,000$g$, the pellets were resuspended in a solution of 18% Dextran/HEPES/HBSS. The solution was centrifuged for 15 min at 3,200$g$, and the pellets were resuspended in a solution of 1% BSA/HBSS/HEPES. The solution was next filtered using a 20-$\mu$m filter, and the filter containing the blood vessel fragments was washed with a 1% BSA solution then centrifuged at 2,000$g$ for 5 min. The pellets containing the blood vessels were stored at −80°C.

## Immunostaining of purified blood vessels

One drop of the solution containing the purified blood vessels (see above) was put on a glass slide with 1 $\mu$l of Cell-Tak (Thermo Fisher Scientific). The sample was fixed in 4% PFA for 15 min at RT. After three washes in PBS, the vessel samples were blocked in a solution of 10% SVF/0.25% Triton X-100/PBS for 30 min. Primary antibodies were diluted in the same solution in which the samples were incubated overnight at 4°C. After PBS washes, the vessels were incubated with the secondary antibodies diluted in 1% SVF/0.1% Triton X-100/PBS for 2 h at RT, followed by an incubation with Hoechst (1:1,500) for 5 min.

## cDNA libraries and RNA sequencing

cDNA library preparation and Illumina sequencing were performed at the École Normale Supérieure genomic core facility (ENS Paris).

PolyA + mRNAs were purified from 300 ng of total RNA using oligo(dT) primers. Libraries were prepared using the strand specific RNA-Seq library preparation TruSeq Stranded mRNA kit (Illumina). They were next multiplexed by six on two high-output flow cells. A 75-bp single read sequencing was performed on a NextSeq 500 (Illumina). A mean of 49 ± 9 million passing Illumina quality filter reads was obtained for each of the 12 samples.

## Bioinformatics

The analyses were performed using the Eoulsan pipeline (Jourdren et al, 2012), including read filtering, mapping, alignment filtering, read quantification, normalization, and differential analysis. Before mapping, poly N read tails were trimmed, reads ≤40 bases were removed, and reads with quality mean ≤ 30 were discarded. Reads were then aligned against the *Mus musculus* genome from Ensembl version 88 using STAR (version 2.7.2d; Dobin et al, 2013). Alignments from reads matching more than once on the reference genome were removed using Java version of Samtools (Li et al, 2009). To compute gene expression, *M. musculus* General Transfer Format (GTF) genome annotation version 88 from Ensembl database was used. All overlapping regions between alignments and referenced exons were counted and aggregated by genes using HTSeq-count 0.5.3 (Anders et al, 2015). The RNA-Seq gene expression data and raw fastq files are available on the GEO repository (www.ncbi.nlm.nih.gov/geo/) under accession numbers: GSE255096 and GSE255097. The sample counts were normalized using DESeq2 1.8.1 (Love et al, 2014). Statistical treatments and differential analyses were also performed using DESeq2 1.8.1.

## Enrichment analysis

Enrichment analysis was conducted with Gene Ontology for BBB and BNB comparison and using Metascape for the oxaliplatin-treated and control mouse BNB comparison.

## qRT-PCR

Total vessel RNAs were extracted using the "Nucleospin RNA XS" kit (Macherey-Nagel) according to the supplier's protocol. Total nerve RNAs were extracted using TRIzol Reagent protocol (Invitrogen). The final RNA concentration (ng.$\mu$l$^{-1}$) was measured using a nanodrop spectrophotometer (Thermo Fisher Scientific) with an absorbance at 260 nm. After cDNA synthesis using superscript III reverse transcriptase (Invitrogen), qRT-PCR was performed with Bio-Rad CFX and associated software for data analysis. *Gapdh*, *Hprt1*, and *Actb* were used as reference genes for relative expression. The list of primers used is summarized in Table 2.

## Western blot analysis

Mouse sciatic nerves were lysed in a protein extraction buffer (50 mM Tris, 150 mM NaCl, 5 mM EDTA, 1% Triton X-100, 1% SDS, 1% protease inhibitor) and homogenized with sterile 5-mm Stainless steel beads, using a Tissuelyser II (QIAGEN) for 2 min at 30 beats per second over two cycles. The samples were sonicated and the

**Table 2. Primers.**

| Gene name | Reference or sequence | Structure/function |
|---|---|---|
| *Actb* encoding β-Actin | QT00095242; QIAGEN | Cytosketelal (housekeeping gene) |
| *Gapdh* | QT01658692; QIAGEN | Metabolic enzyme (housekeeping gene) |
| *Hprt1* | QT00166768; QIAGEN | Metabolic enzyme (housekeeping gene) |
| *Pecam-1* | F GCACCCATCACTTACCACCT | Endothelial cells (EC) |
| | R GCTCGTCCCCTCTTTCACA | |
| *Cldn5* | F TAAGGCACGGGTAGCACTCA | Tight junction between EC |
| | R GGACAACGATGTTGGCGAAC | |
| *Zo-1* | F GTTGGTACGGTGCCCTGAAAGA | Tight junction between EC |
| | R GCTGACAGGTAGGACAGACGAT | |
| *Sm22a* | F CAACAAGGGTCCATCCTACGG | Adult smooth muscle |
| | R ATCTGGGCGGCCTACATCA | |
| *Mpz* | F AAGAACATGATGGGCCTGGA | Myelin sheath |
| | R TGAGGAGCAAGAGGAAAGCA | |
| *Prx* | F TGAAGCTACCCACCCTCAAG | Schwann cells |
| | R CTGACATTTTGGGCAGCTGT | |
| *Tuj1* | F AGCGCATCAGCGTATACTACAA | Neuron cytoskeletal |
| | R TAAAGTTGTCGGGCCTGAATAG | |
| *Gfap* | F CACCTACAGGAAATTGCTGGAGG | Non-myelinating Schwann cells |
| | R CCACGATGTTCCTCTTGAGGTG | |
| *S100 β* | F CTGGAGAAGGCCATGGTTGC | Schwann cells |
| | R CTCCAGGAAGTGAGAGAGCT | |
| *Acta2* | Bio-Rad qMmuCID0006375 | Smooth muscle cells |
| *Myh11* | Bio-Rad qMmuCID0019272 | Smooth muscle cells |
| *Mylk* | Bio-Rad qMmuCID0005600 | Smooth muscle cells |
| *Ryr3* | Bio-Rad qMmuCED0050099 | Skeletal muscle contraction |
| *Vegfr1* | QT00096292; QIAGEN | Vascular endothelial growth factor receptor 1 |
| *Vegfa* | QT00160769; QIAGEN | Vascular endothelial growth factor A |

supernatants obtained after centrifugation (20,800*g* for 20 min at 4°C) were used as samples. The extracted proteins were assayed with the Pierce BCA Protein Assay Kit (Life technologies) by spectrophotometry using BSA as a standard. Equal quantities of proteins (20 μg) were denatured by heating in Laemmli buffer, DTT, then separated in a Mini-PROTEAN TGX Stain Free Gels 4–15% (Bio-Rad). The proteins were then transferred to a 0.2 μm PVDF membrane (Bio-Rad). The non-specific binding sites of the membrane were blocked in Tris buffer containing 1% Tween (TBS-T) supplemented with 5% milk powder, for 1 h at RT. The membrane was next incubated in TBS-T supplemented with 5% milk powder, mixed with anti-VEGF-A, anti-HIF-1α, or anti-H3 overnight at 4°C. After washing, it was incubated with antibodies conjugated with HRP peroxidase (1/5,000) for 1 h at RT. The revelation was made by chemiluminescence with the Supersignal ECL Plus or Femto kit (Thermo Fisher Scientific). The signals obtained were acquired on Fusion FX system. The level of chemiluminescence for each antibody was normalized to histone-H3 signal. HIF-1α single band was observed around 55 kD in the sciatic nerve (Fig 7C) whereas with the same

protocol a band around 100 kD is observed in brain protein extract (data not shown).

### Platinum concentration detection via mass spectrometry

Mice were euthanized by intraperitoneal (i.p.) injection of 400 mg/kg ketamine (Imalgene) and 20 mg/kg Xylazine (Rompun). Blood was collected directly from the heart on EDTA tubes (Microvette, Sarstedt), followed by intracardiac perfusion with PBS1X +/+ (Life Technologies) over 5 min until exsanguination. Plasma was centrifuged for 10 min at 6,800*g* and 4°C, then stored at −80°C. After perfusion, sciatic nerves and DRG were dissected and frozen directly in liquid nitrogen, then stored at −80°C. Nerves and DRGs were cold-ground with sterile 5-mm Stainless steel beads, using a Tissuelyser II (QIAGEN) for 2 min at 30 beats per second over two cycles. Homogenates were resuspended in 500 μl ultrapure water and freeze-dried by vacuum evaporation for 2 h at 42°C. Eppendorf microtubes containing dry samples (sciatic nerves and DRGs) were rinsed with 10% $HNO_3$ (vol/vol), and the solutions were transferred

in Teflon beakers and evaporated at 80°C. In each sample, 1 ml of pure $HNO_3$ (67% vol/vol) was added to the residue, and closed Teflon beakers were place overnight on a hot plate at 100°C. Finally, the solutions were evaporated at 80°C and the residue dissolved in 10 ml of 1% $HNO_3$ (vol/vol) after 20 min in ultrasonic bath and 1 h on a hot plate at 100°C. To prepare the plasma samples for Platinum analysis with ICP-MS, a simple 1,000-fold dilution was made with a mixture of 1.5% TMAH (tetramethyl ammonium hydroxide) to dissolve the protein-rich liquids and 1% $HNO_3$. To check potential contaminations (reagents, vials...) during sample preparation, procedural blanks were prepared following the same protocols than the samples. Platinum concentrations were determined with the Thermo Fisher Scientific iCAP TQ ICP-MS (using the Kinetic Energy Discrimination mode and He as collision gas) at the "Plateforme AETE-ISO, OSU OREME, Université de Montpellier-France." An internal solution was added on-line to the samples to correct signal drifts. A calibration curve including four points (0, 1, 5, and 10 ppb) was analyzed every 20–30 samples. To check the accuracy of the results, sciatic nerve, DRG and plasma samples from sham control animals were doped with oxaliplatin, homogenized, prepared following the same protocol than the samples from treated animals and analyzed. The deviation from the theoretical value was less than 10% and the analytical error (relative SD) less than 5%. Procedural blank contribution for plasma and solid samples is less than 4% and 0.2%, respectively.

## Statistical analysis

Data shown on graphs are expressed as means and SEM. Graphs and statistical analyses were performed using R Studio and Graphpad Prism software. The different statistical tests used were specified in the Figure legends. Volcano plots were generated using Enhanced Volcano (https://bioconductor.org/packages/devel/bioc/vignettes/EnhancedVolcano/inst/doc/EnhancedVolcano.html) and the heatmap was performed using Superheat (Barter & Yu 2018) with R Studio.

# Supplementary Information

# Acknowledgements

We thank Drs Nathalie Kubis, Pierre Lozeron, and Marc Pocard for valuable comments throughout the course of this study and insightful discussion on its clinical relevance. We thank Salvatore Cisternino, Camille Cotteret, and Camille Gons who provided us with Oxaliplatin. We also deeply thank all current and previous members of the Brunet laboratory for helpful comments and discussion on the project. We thank the ORION technological Core (IMACHEM-IBiSA) at the CIRB-Collège de France for their assistance in image acquisition and scripts. We thank the Genomique ENS core facility, supported by the France Génomique national infrastructure, funded as part of the "Investissements d'Avenir" program managed by the Agence Nationale de la Recherche (contract ANR-10-INBS-0009). We thank the AETE-ISO platform, OSU-OREME/Université de Montpellier for the mass spectrometry analysis of the platinum concentration in our samples. This work was

supported by research grants from INSERM CoPoC NEUROPLAT, Fondation Collège de France NEUROCURE, the French "League against Cancer," and annual laboratory funding of I Brunet. Over the course of this project, S Taïb was the recipient of fellowships from the "Ministère de l'Enseignement Supérieur et de la Recherche" (MESR), "Labex MemoLife/PSL University," and the French "League against Cancer". J Durand received a salary from a French National Research Agency (ANR) funding attributed to I Brunet and a fellowship from the ARC Cancer Foundation. A-C Boulay was supported by postdoctoral fundings from Fondation pour la Recherche sur la Sclérose En Plaques (ARSEP) and M Cohen-Salmon by FRM. V Dehais is a recipient of a PhD fellowship from the MESR.

## Author Contributions

S Taïb: conceptualization, data curation, formal analysis, validation, investigation, visualization, and methodology.
J Durand: conceptualization, data curation, software, formal analysis, validation, investigation, visualization, methodology, and writing—original draft.
V Dehais: formal analysis, validation, investigation, visualization, and methodology.
A-C Boulay: formal analysis, investigation, and methodology.
S Martin: resources, formal analysis, validation, investigation, visualization, and methodology.
C Blugeon: data curation and methodology.
L Jourdren: data curation, formal analysis, and methodology.
R Freydier: data curation, formal analysis, investigation, and methodology.
M Cohen-Salmon: methodology.
J Hazan: conceptualization, formal analysis, and writing—original draft, review, and editing.
I Brunet: conceptualization, resources, supervision, funding acquisition, validation, visualization, methodology, project administration, and writing—original draft, review, and editing.

## Conflict of Interest Statement

The authors declare that they have no conflict of interest.

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
