## [Reviewer comments · Life Science Alliance]

Life Science Alliance

Vascular dysfunction is at the onset of oxaliplatin-induced peripheral neuropathy symptoms in mice

Isabelle Brunet, Sonia Taib, Juliette Durand, Vianney Dehais, Anne-Cécile Boulay, Sabrina Martin, Corinne Blugeon, Laurent Jourden, Rémi Freydier, Martine Cohen-Salmon, and Jamilé Hazan

DOI: <https://doi.org/10.26508/lsa.202402791>

Corresponding author(s): *Isabelle Brunet, Centre Interdisciplinaire de Recherche en Biologie*

Review Timeline:

Submission Date:	2024-04-23
Editorial Decision:	2024-05-30
Revision Received:	2024-10-29
Editorial Decision:	2024-11-01
Revision Received:	2024-11-12
Accepted:	2024-11-12

Scientific Editor: *Eric Sawey, PhD*

Transaction Report:

May 30, 2024

Re: Life Science Alliance manuscript #LSA-2024-02791-T

Dr. Isabelle Brunet
CIRB Collège de France/CNRS UMR 7241/INSERM U1050
11 Place Marcelin Berthelot
Paris 75005

Dear Dr. Brunet,

Thank you for submitting your manuscript entitled "Vascular dysfunction is at the onset of oxaliplatin-induced peripheral neuropathy symptoms in mice" to Life Science Alliance. The manuscript was assessed by expert reviewers, whose comments are appended to this letter. We invite you to submit a revised manuscript addressing the Reviewer comments. Please do not remove the OIPN data in response to Reviewer 1's comments, but instead address the dosage concerns.

Thank you for this interesting contribution to Life Science Alliance. We are looking forward to receiving your revised manuscript.

Sincerely,

B. MANUSCRIPT ORGANIZATION AND FORMATTING:

Reviewer #1 (Comments to the Authors (Required)):

The study presented by Taïb et al., aims at characterizing the vascular contribution to oxaliplatin-induced peripheral neuropathy (OIPN). Using a mouse model of OIPN, authors first compared transcriptomic analyses from purified brain and sciatic nerve blood vessels and nerve blood vessels after oxaliplatin and control administration. The molecular profiling of purified sciatic nerve and brain blood vessels revealed 469 genes highly enriched in snBV and 502 highly enriched in brBV. Using IHC, specific snBV markers that selectively label the distinct compartments or blood vessel types composing the vasa nervorum were identified. Several differences in oxaliplatin-transporter expression were found and could partly account for the different drug permeability of the two barriers, as well as for the selective alteration of the peripheral nervous system in oxaliplatin-induced peripheral neuropathy. Authors then used oxaliplatin, intravenously injected (cumulative dose of 30 mg/kg), to explore its impact on the vasa nervorum. Subsequent results are matter of discussion and raise several concerns.

Major points:

- How do authors explain that a single injection of oxaliplatin at such a high dose did not induce hypersensitivity (Figure 4B)? A single 2 mg/kg (iv) is sufficient to induce acute OIPN symptoms in the literature. Conversely, it did induce OIPN symptoms in figure 6C tactile allodynia, but gave inconsistent results on cold sensitivity after inj1 or 2 (Figure 6D). Baseline are also very different regarding cold paw withdrawal thresholds between figure 4 (around 5 sec) and figure 6 (around 16 sec), or distinct behavior have been examined? And so why providing different readout for cold hypersensitivity?
 - I strongly suggest to eliminate the oxaliplatin part (OIPN model) of this work or to use more "relevant pharmacological" dose oxaliplatin and evaluate its short term effect on vascular function. Hence, the technical part comparing the distinct compartments or blood vessel types composing the vasa nervorum is of concern. But, due to the high dose oxaliplatin used and to inconsistent behavioral results, one can not conclude that the vascular dysfunction is at the onset of oxaliplatin-induced peripheral neuropathy symptoms in mice. It can be a consequence of repeated high dose oxaliplatin injections, but not at the onset of OIPN.
 - the title should be changed to better reflect the results and conclusions that can be drawn from this study.
- Minor (maybe major) point: Have the behavioral tests been done blinded to treatments?

Reviewer #2 (Comments to the Authors (Required)):

This is an interesting paper about the role of vessel response in oxaliplatin- induced neuropathy. Minor changes could improve the ms.

- Why the authors chose a protocol of 3 oxaliplatin administration? it is not close to the clinical setting
- When pain measures were done after tadalafil or ambrisentan administration (1 h? 30 min? 24 h?)
- The role of VEGF should be better evaluated since the main role of this factor in pain induction, please see and discuss doi: 10.1186/s13046-021-02127-x and doi: 10.1016/j.jccell.2015.04.017.

Anyway the manuscript is well conceived and offers really interesting data. My compliments

Reviewer #3 (Comments to the Authors (Required)):

The manuscript presented by Taïb et al. addresses an unmet need in oncology, by explaining and preventing acute oxaliplatin-induced peripheral neuropathy.

The main results is that OIPN would be the consequences of an acute ischemia of peripheral nerves, and that the prevention of this ischemia would prevent OIPN. This result is innovative (at least for me).

The manuscript is well written and quite clear. The method and the results are very interesting. However, I have several interrogations regarding this new explanation of the acute OIPN pathophysiology.

Here are my comments:

Introduction: no comment.

Results

It is surprising that the gene *abcb1* (P-glycoprotein) is not underlined in the molecular signature of snBV and brBV (Figure 3B). Since P-gp seems to be a dominant factor of the BBB integrity and function (and probably BNB).

Similarly, OCT2 gene seems to be not involved in the snBV, whereas OCT2 has been described as important factor leading to oxaliplatin-related CIPN (neurotoxicity), <https://pubmed.ncbi.nlm.nih.gov/36506732/> and <https://pubmed.ncbi.nlm.nih.gov/23776246/>. (Neuron level?)

Figure EV3, could you specify the timing for this analysis (which day after the beginning of oxaliplatin injection?)
It is possible that the oxaliplatin treatment is too short to highlight a decrease in the IENFD.

Figure 5C, I am not sure that the representation of OCT, OCTN, and CTR1 is correct. We can understand that these transporters help for the crossing of endothelial cells. I believe that these transporters help to cross the membrane and not the cell (i.e., to enter into the endothelial cell). And for example, it is highly possible that MATE help to go out the endothelial cells, to reach neuron and glial cells (such as it is described in kidney for the nephrotoxicity of cisplatin, cisplatin is a good substrate of OCT transporters but not for MATE, inducing an accumulation of platinum in endothelial cells in the kidneys).

Figure 7: nice! (reading the manuscript, I was thinking to do such analysis...)

Discussion

Could you please provide a reference for "However, targeting OCT2 could prove inapplicable from a therapeutic point of view, as altering oxaliplatin transport could reduce its antitumoral efficacy."

I do not understand the meaning of this sentence, in the context of the present study. "So far, mouse models of acute OIPN have all been developed by single or repeated intraperitoneal injection(s) of oxaliplatin, which exclude to evaluate direct effect of oxaliplatin onto EC and mural cells of the vasa nervorum (Gauchan et al., 2009; Descoeur et al., 2011; Ogihara et al., 2019; Gould et al., 2021; Wang et al., 2023)." Do you mean that with an intraperitoneal injection, oxaliplatin should have no effect on the vasa nervorum? If yes, why? After an IP injections, oxaliplatin reaches the general blood circulation and nerves, since following IP injections, oxaliplatin induces neurotoxicity.

For each figure, take care to explain each abbreviation of the figures (it is sometimes hard to understand their meaning).

Could you explain the choice of each drug dose?

Could you explain the choice of each gene/protein as biomarker of each structure/function assessed? There are a lot of gene/protein assessed and it would help the reader to repeat some works)

Using ambrisentan and/or tadalafil, would it be possible that these drugs increase the neurotoxicity of oxaliplatin, by increased the blood flow in nerves, and consequently the nerve/neuron exposures? Moreover, some alternative strategies to prevent OIPN propose to reduce the blood flow in extremities to prevent CIPN, such as cryotherapy (<https://pubmed.ncbi.nlm.nih.gov/32955997/>) or compression (<https://pubmed.ncbi.nlm.nih.gov/38060077/>) of extremities. Do you have some explanations of such opposite strategies in comparison to your results? Does ambrisentan and/or tadalafil may prevent chronic OIPN?

In some way, the first part of the work on transporters is not very useful for the final results. However, these explorations rise the question of a possible interaction between ambrisentan and/or tadalafil and oxaliplatin (blocking transporters involved in the oxaliplatin influx or efflux)? Is there any assessment of platinum exposure of DRG and nerves after ambrisentan and/or tadalafil treatment?

Finally, may ambrisentan and/or tadalafil have analgesic effect in control animals? Or do you have some arguments that it is not the case?

Reviewer #1 (Comments to the Authors (Required)):

The study presented by Taïb et al., aims at characterizing the vascular contribution to oxaliplatin-induced peripheral neuropathy (OIPN). Using a mouse model of OIPN, authors first compared transcriptomic analyses from purified brain and sciatic nerve blood vessels and nerve blood vessels after oxaliplatin and control administration. The molecular profiling of purified sciatic nerve and brain blood vessels revealed 469 genes highly enriched in snBV and 502 highly enriched in brBV. Using IHC, specific snBV markers that selectively label the distinct compartments or blood vessel types composing the vasa nervorum were identified. Several differences in oxaliplatin-transporter expression were found and could partly account for the different drug permeability of the two barriers, as well as for the selective alteration of the peripheral nervous system in oxaliplatin-induced peripheral neuropathy. Authors then used oxaliplatin, intravenously injected (cumulative dose of 30 mg/kg), to explore its impact on the vasa nervorum. Subsequent results are matter of discussion and raise several concerns. Major points:

- How do authors explain that a single injection of oxaliplatin at such a high dose did not induce hypersensitivity (Figure 4B)? A single 2 mg/kg (iv) is sufficient to induce acute OIPN symptoms in the literature. Conversely, it did induce OIPN symptoms in figure 6C tactile allodynia, but gave inconsistent results on cold sensitivity after inj1 or 2 (Figure 6D). Baseline are also very different regarding cold paw withdrawal thresholds between figure 4 (around 5 sec) and figure 6 (around 16 sec), or distinct behavior have been examined? And so why providing different readout for cold hypersensitivity?

We thank the Referee for giving us the opportunity to clarify this point.

First, it should be pointed that our main goal was here to provide an intravenously injected acute OIPN mouse model, which showed acquired neuropathic symptoms at the end of the protocol as assayed by behavioral tests. This protocol ends with the mouse sacrifice and the beginning of our anatomical, cellular and transcriptomic analyses, hence it was critical for us that all injected mice exhibited clear and significant symptoms at that stage. Therefore, the fact that tactile allodynia occurred after the first or second injection of oxaliplatin was not critical for our paradigm. Nevertheless, as pointed by the Referee, a single injection of oxaliplatin at 10 mg/kg (see below extract from **Fig 4B**) led to a strong tendency (VF $p=0.075$) for allodynia, while this symptom is already significant after 1 injection in **Fig 6C** (see below extract). The difference in significance is here mainly due to the different number of tested animals and thus to the statistical power of the analysis ($N=5$ mice/group for **Fig 4** versus $N=11-12$ mice/group for **Fig 6**). Furthermore, in our hands, a single injection of oxaliplatin clearly resulted in an intermediate situation where differences in sensitivity could still be observed between mouse groups. These differences may be partially explained by the fact that mice, as humans, might not develop symptoms immediately after injection but with a slight delay that can vary from an animal to the other. This reinforces the rationale of having an appropriate number of injections to generate an acute OIPN mouse model, which invariably shows robust and significant neuropathic symptoms.

Regarding the dose of oxaliplatin, almost each OIPN mouse model of the recent literature uses a different dose of oxaliplatin, from a single injection of 2 mg/kg (though via pressurized regional IV; Shankara Narayan et al., *Discov Oncol* 2022 13(1):21) to a cumulative dose of 72 mg/kg (Chiorazzi et al., *Eur J Pharmacol* 2018 840:89-103), yet with an average

cumulative dose of 10-25 mg/kg in the majority of analyses including behavioral assays (e.g., Pereira et al., *Neurotox Res.* 2021 39(6):1782-99; Dziubina et al., *Int J Mol Sci* 2022 23(7):4057; Arias et al., *Anest Analg* 2023 137(3):691-701; Maia et al., *Neuropharmacol* 2024 245:109828). Furthermore, it has been shown that the impact of a given oxaliplatin dose varies with the sex, age and genetic background of the animal, while the outcome measures in the assessment of OIPN also condition the dose to be tested (Warncke et al., *Front Pain Res* 2021 2:683168). Hence, although we acknowledge that *a single 2 mg/kg (iv)* is supposedly *sufficient to induce acute OIPN symptoms*, we based our initial protocol/dose (several injections of 10 mg/kg) on the study by Ogihara et al. (*J Pharmacol Sci* 2019 141(4):131-8), which invariably showed significant OIPN hypersensitivity symptoms. Nevertheless, to reach a conclusion on a relevant oxaliplatin dose for our analysis, we tested several iv doses 2, 5 and 10 mg/kg but failed to replicate the hypersensitivity/neuropathic symptoms of our OIPN mice with a dose of 2 mg/kg (even after 3 injections). Indeed, in our hands and with our behavioral tests, only 5 and 10 mg/kg doses led to robust and significant neuropathic symptoms, although **not after a single injection** (see below and in Figs 4 & 6).

(from Fig S2)

(from Fig 4)

(from Fig 6)

As for the legitimate referee's criticism on the cold paw withdrawal test thresholds that indeed differ between Figure 4 (around 5 sec) and Figure 6 (around 16 sec), the reason is simply that distinct behaviors, yet both investigating cold hypersensitivity, had been explored due to ethical concerns and limitations. This was described in the *Material and Methods* section of the revised manuscript (Cold Plate test, p. 32). The initial method (Fig 4), involving exposure of the mouse paw to a painful stimulus for two minutes at 4°C without any possibility for withdrawal was deemed unethical according to the European/French new regulations for animal handling, as it might cause significant suffering even for control animals. Although this method was initially ethically approved and used to develop our mouse model, it could not be continued. In contrast, the method depicted in Fig 6, which allows the temperature to decrease at a rate of 2°C per minute, defines a different threshold at which the first response of the animal was recorded. This alternative approach was considered less sensitive/painful and aligns with our recent ethical guidelines, which prioritize the animal welfare while still aiming at gathering robust data. As shown in Fig 6, at the end of this new protocol (i.e., only after the third injection, see point above), we observed a significant cold hypersensitivity. This could be stated in the text and/or figure legend if requested.

-I strongly suggest to eliminate the oxaliplatin part (OIPN model) of this work or to use more "relevant pharmacological" dose oxaliplatin and evaluate its short term effect on vascular function. Hence, the technical part comparing the distinct compartments or blood vessel types composing the vasa nervorum is of concern. But, due to the high dose oxaliplatin used and to inconsistent behavioral results, one can not conclude that the

vascular dysfunction is at the onset of oxaliplatin-induced peripheral neuropathy symptoms in mice. It can be a consequence of repeated high dose oxaliplatin injections, but not at the onset of OIPN.

In the revised version of our manuscript and in this response (see above), we provided additional data showing that a lower dose of oxaliplatin **at 5 mg/kg** (yet not at 2 mg/kg) induced similar neuropathic symptoms that are prevented by tadalafil administration. We also showed that mice regained normal sensitivity after discontinuation of oxaliplatin (Fig S1) despite the use of high doses and return to normal weight (data not shown), validating the acute nature and the relevance of our OIPN model. We thus believe that the main take-home message of our article remains the rescue of oxaliplatin-induced neuropathic symptoms and the hypoxia-reflecting increase in HIF-1 α expression within the nerve (quantified by both immunostaining and western blot analyses) with vasodilators.

Moreover, we here would like to stress that **the so-called inconsistency of our behavioral results** is due to the number of animals included in the Von Frey test for the difference of significance after the first or second injection, and to a different cold plate test paradigm dictated by new regulations of French ethical committees (as discussed above). Our data are thereby by no means inconsistent, whereas all our results are significant at the end of our protocol, even with the less stringent cold plate paradigm of Fig 6. It should also be noted that we did not identify any significant increase in any apoptosis pathway that could reflect toxic high dose effect. Conversely, our comparative transcriptomic analysis highlighted an increased expression of genes involved in vascular smooth muscle contraction pathway, reflecting vasoconstriction occurring upon oxaliplatin administration, which further validates our conclusions. Finally, the observation that **vasodilators significantly diminished or prevented both the neuropathic symptoms and nerve hypoxia of oxaliplatin-injected animals** does not argue for a global toxic effect of oxaliplatin high doses. Furthermore, the fact that **similar conclusions could be drawn at a lower dose of oxaliplatin (5 mg/kg)** for which both neuropathic symptoms and increased HIF-1 α expression were prevented by tadalafil co-administration (see extracts below from Fig S2 for behavioral tests and Fig 7C for WB), conclusively established the involvement of vascular dysfunction. We believe that this fully addresses the reviewer concern.

-the title should be changed to better reflect the results and conclusions that can be drawn from this study.

We do not consider changing the title as we believe that it fully reflects our data as discussed above: in our model, vascular vasoconstriction leads to nerve hypoxia as reflected by increased HIF-1 α and VEGFA, which participates to the neuropathic symptom appearance. Preventing the occurrence of this vasoconstriction led to the drastic reduction or the actual suppression of neuropathic symptoms, which suggests that this vasoconstriction is an early event in the neuropathic process.

Minor (maybe major) point: Have the behavioral tests been done blinded to treatments?

We aimed to blindly perform the behavioral tests but this was not always possible due to technical constraints, and mostly because any experimenter could recognize the oxaliplatin-treated mice. Nevertheless, the experimenter would not know which group was treated with oxaliplatin alone or in association with vasodilators. Furthermore, over the course of the study, at least two members of the lab performed the behavioral tests, and two authors blindly analyzed the behavioral results, while the same conclusion could be drawn from each analysis.

Reviewer #2 (Comments to the Authors (Required)):

This is an interesting paper about the role of vessel response in oxaliplatin- induced neuropathy. Minor changes could improve the ms.

- Why the authors chose a protocol of 3 oxaliplatin administration? it is not close to the clinical setting

We here aimed at generating an acquired acute OIPN mouse model to unravel the behavioral and molecular phenotype with a special emphasis on changes occurring at the initiation of symptoms. Intravenous (Iv) injection mimics the mode of administration to patients more accurately than intraperitoneal injection (Ip). In addition, since Ip and local intramuscular (Im) injections of oxaliplatin do not result in equivalent changes in motor and sensory nerve excitability (Makker et al., J Neurophysiol 2020 124(1):232-44), it appeared critical to use Iv administration to properly investigate the vascular component of OIPN.

Given the high metabolic rate in mice and the hypersensitivity reaction to oxaliplatin injection, we carefully chose this administration protocol to combine reliably acquired neuropathic symptoms with a timeframe corresponding to the start of the pathology. We also assessed that this model recapitulated the reversible nature of neuropathic symptoms as defined for an acute OIPN model. Furthermore, to minimize high-dose toxicity (see response to Referee1) while ensuring robust behavioral symptoms at the end of the protocol, we tested different doses, which defined a most suited administration of 3 x 10 mg/kg. In our hand, this approach balanced significant data for all mouse groups tested, reversible symptoms and the desired therapeutic outcomes. We observed that this protocol provides an optimized window for robustly assessing the behavioral and molecular changes triggered by oxaliplatin

administration. However, to address the comments of Reviewer 1 about the high dose toxic effect (i.e., a cumulative dose of 30 mg/kg), we undertook the same experiments with a 5-mg/kg dose of oxaliplatin, which was equally efficient to monitor both behavioral and molecular alterations due oxaliplatin administration (see extracts from Fig S2 and Fig7), and that may be appropriately used to generate a chronic OIPN model in the future.

- When pain measures were done after tadalafil or ambrisentan administration (1 h? 30 min? 24 h?)

Ambrisentan and tadalafil have respective half-lives of 15 and 17.5 hours in humans. Von Frey tests were accordingly performed between 30 minutes and 2 hours after ambrisentan injections and the day after tadalafil injections. The cold plate test was performed 5 to 10 hours after the injection of ambrisentan or tadalafil.

- The role of VEGF should be better evaluated since the main role of this factor in pain induction, please see and discuss doi: 10.1186/s13046-021-02127-x and doi: 10.1016/j.ccell.2015.04.017.

These two papers demonstrated the pro-nociceptive effect of VEGF-A-induced VEGFR-1 activation in peripheral sensory neurons in both cancer pain and oxaliplatin-induced peripheral neuropathy. In cancer, tumor-derived VEGF-A activates VEGFR-1 signaling in mouse peripheral sensory neurons (Selvaraj et al., *Cancer Cell* 2015 27(6):780-96), while increased astrocyte-derived VEGF-A evokes neuropathic pain via the activation of VEGFR1 signaling in the nociceptive neurons of the spinal cord dorsal horn in OIPN as well as in paclitaxel- and vincristine-induced neuropathy (Micheli et al., *J Exp Clin Cancer Res* 2021 40(1):320). Given that VEGF genes are downstream targets of HIF-1 α (Liu et al., *Circ Res* 1995 77:638-43; Forsythe et al., *Mol Cell Biol* 1996 16:4604-13), it is indeed of interest to assess the involvement of the VEGF pathway in our acute OIPN mouse model and to determine the impact of vasodilator administration on this signaling.

To this end, we performed RT-qPCR, western blot and immunohistochemical analyses for two key cues of VEGF signaling, VEGF-A and VEGFR-1. Notably, we observed that the levels of VEGF-A mRNA and protein were significantly increased in entire oxaliplatin-treated sciatic nerves (Fig 7 D-E, see extracts below), as expected for HIF-1 α -overexpressing hypoxic nerves (Fig 7 A-B & Liu et al., *Circ Res* 1995 77:638-43). Moreover, immunostaining analysis showed that VEGFR-1 was significantly increased in oxaliplatin-treated endoneurial vessels and that this upregulation was rescued by tadalafil administration (Fig S6, see below).

Interestingly, Kiguchi et al. (*J Neurochem* 2014 129(1):169-78) showed that the hypoxia triggered by sciatic nerve ligation induced an increase in VEGF-A in the sciatic nerve, which is mediated by the recruitment of bone marrow macrophages and neutrophils. Therefore, its observed accumulation in oxaliplatin-treated sciatic nerves may not result from astrocytes as in Micheli et al. (*J Exp Clin Cancer Res* 2021 40(1):320), but might originated from circulating immune cells. In our study, macrophages could similarly activate the VEGF pathway and induce neuro-inflammation, possibly leading to chronic stages of peripheral neuropathy.

Furthermore, the fact that tadalafil administration prevents the recruitment of VEGFR-1 to intra-nervous blood vessels (Fig S6) further supports our hypothesis stating that oxaliplatin-

induced nerve hypoxia, caused by increased vasoconstriction of nerve blood vessels, plays a key role at the onset of OIPN, most likely through the VEGF pathway. We would like to thank the referee for prompting us to address the involvement of this signaling.

(Fig 7)

(Fig S6)

Anyway the manuscript is well conceived and offers really interesting data. My compliments

Reviewer #3 (Comments to the Authors (Required)):

The manuscript presented by Taïb et al. addresses an unmet need in oncology, by explaining and preventing acute oxaliplatin-induced peripheral neuropathy.

The main results is that OIPN would be the consequences of an acute ischemia of peripheral nerves, and that the prevention of this ischemia would prevent OIPN. This result is innovative (at least for me).

The manuscript is well written and quite clear. The method and the results are very interesting. However, I have several interrogations regarding this new explanation of the acute OIPN pathophysiology.

Here are my comments:

Introduction: no comment.

Results

It is surprising that the gene *abcb1* (P-glycoprotein) is not underlined in the molecular signature of snBV and brBV (Figure 3B). Since P-gp seems to be a dominant factor of the BBB integrity and function (and probably BNB).

We thank the reviewer for this point. BBB and BNB share similar molecular characteristics (e.g., *Tjp1/ZO-1*) but also display critical differences (e.g., *Unc5b*) as supported by our transcriptomic analysis. Yet, we have indeed observed that *Abcb1* genes, namely *Abcb1a* and *Abcb1b* in rodents, are commonly expressed in the two vasculatures. As they play a key role in the permeability of the BBB, so as *Abcg2*, we totally agree with the referee that it is of high interest to investigate their functional contribution to the alteration of the BNB in the context of OIPN. For now, we have just added *Abcb1a* and *Abcb1b* to the heatmap representation of sciatic nerve and brain blood vessel commonly and differentially expressed genes Fig. 3B and in the text (p. 6).

Similarly, OCT2 gene seems to be not involved in the snBV, whereas OCT2 has been described as important factor leading to oxaliplatin-related CIPN (neurotoxicity), <https://pubmed.ncbi.nlm.nih.gov/36506732/> and <https://pubmed.ncbi.nlm.nih.gov/23776246/>. (Neuron level?)

Several papers have indeed highlighted the key role of the OCT2 transporter in the internalization of oxaliplatin into dorsal root ganglion neurons (e.g., Huang et al., J Clin Invest 2020 130(9): 4601-6). Furthermore, its inhibition tends to prevent neuropathic pain induced by oxaliplatin without altering its anticancer effect (Nepal et al. Cancer Res Comm 2022 2(11):1334-43). All these articles however address the involvement of OCT2 in oxaliplatin **neuronal uptake**, but to our knowledge no oxaliplatin transporters have been specifically defined for brain or sciatic nerve blood vessels. Indeed, our transcriptomic analysis (see Fig 3B & 5D) showed a similarly low expression of OCT2-encoding transcripts in both brain and sciatic nerve **blood vessels**, which does not imply that OCT2 is absent from the *vasa nervorum* nor that it has no critical role in the initiation of OIPN in our model. Yet, oxaliplatin accumulates within the sciatic nerves of our OIPN-treated mice, although less strikingly than in the DRGs (Fig 6E), but through which uptake mechanism remains to be determined.

Figure EV3, could you specify the timing for this analysis (which day after the beginning of oxaliplatin injection?)

It is possible that the oxaliplatin treatment is too short to highlight a decrease in the IENFD.

All post-mortem samples were analyzed at the end of the injection protocol, the day after the third injection of oxaliplatin (i.e., 5 days after the first injection). On the day of euthanasia, the animals still showed neuropathic symptoms (Fig 4) without IENF loss (Fig S4). We have also established that the neuropathic symptoms induced by oxaliplatin in this model

are reversible around 30 days after the first oxaliplatin injection (Fig S1). This further suggests that the behavioral alterations of this mouse model are not irreversible. Thus, we agree with the referee that it might be too early to see a loss of IENFD, which is an additional argument for stating that our model is at the acute phase of the neuropathy.

Figure 5C, I am not sure that the representation of OCT, OCTN, and CTR1 is correct. We can understand that these transporters help for the crossing of endothelial cells. I believe that these transporters help to cross the membrane and not the cell (i.e., to enter into the endothelial cell). And for example, it is highly possible that MATE help to go out the endothelial cells, to reach neuron and glial cells (such as it is described in kidney for the nephrotoxicity of cisplatin, cisplatin is a good substrate of OCT transporters but not for MATE, inducing an accumulation of platinum in endothelial cells in the kidneys).

We fully agree with this comment as we do not know the exact cellular distribution of oxaliplatin transporters at the *vasa nervorum*. We have shown that transporters of the OCT family, CTR1 and MATE1 are expressed in intra-nervous blood vessels through our transcriptomic analysis. They may indeed mediate oxaliplatin accumulation in endothelial cells, as well as its efflux into the nerve fascicles or into the blood, and thereby participate in oxaliplatin neurotoxicity. We have thus modified our representation (Fig 5C) to more comprehensively specify the potential locations of these transporters.

Figure 7: nice! (reading the manuscript, I was thinking to do such analysis...)

Discussion

Could you please provide a reference for "However, targeting OCT2 could prove inapplicable from a therapeutic point of view, as altering oxaliplatin transport could reduce its antitumoral efficacy."

Again, we thank the referee for commenting on this unclear shortcut. One reference describing a therapeutic strategy aiming at preventing OIPN through the targeting of OCT2 in DRG neurons with duloxetine, which showed a potent OCT2-inhibitory activity is the following: Nepal et al. *Cancer Res Comm* 2022 2(11):1334-43. Yet, if this study suggests that inhibition of OCT2 counteracts oxaliplatin accumulation in the DRGs without affecting the anti-tumor properties of oxaliplatin in mice, other studies propose combinations of molecules to activate OCT2 as a therapeutic option for sensitizing cancer cells to oxaliplatin to improve cancer treatment (e.g., Liu et al., *Sci Transl Med* 2016 8(348): 348ra97). We thus believe that the strategy aiming at targeting OCT2 to simultaneously prevent OIPN and increase oxaliplatin efficiency in cancer treatment might be difficult to implement. We have modified this sentence in the Discussion accordingly.

I do not understand the meaning of this sentence, in the context of the present study. "So far, mouse models of acute OIPN have all been developed by single or repeated intraperitoneal injection(s) of oxaliplatin, which exclude to evaluate direct effect of oxaliplatin onto EC and mural cells of the vasa nervorum (Gauchan et al., 2009; Descoeur et al., 2011; Ogihara et al., 2019; Gould et al., 2021; Wang et al., 2023)." Do you mean that with an intraperitoneal

injection, oxaliplatin should have no effect on the vasa nervorum? If yes, why? After an IP injections, oxaliplatin reaches the general blood circulation and nerves, since following IP injections, oxaliplatin induces neurotoxicity.

There again we totally agree with the referee as the term 'exclude' is far too strong, but we here initially aimed at mimicking the mode of administration to patients. Yet, we do indeed expect that both IP and IV injections of oxaliplatin have an impact on the *vasa nervorum* although this effect might slightly vary due to several considerations. IP injections will surely enable oxaliplatin to reach the general blood circulation and nerves, although its concentration and reactivity may differ from those observed after an IV injection. It was for instance shown that IP and local intramuscular injections of oxaliplatin do not lead to similar changes in motor and sensory nerve excitability (Makker et al., J Neurophysiol 2020 124(1):232-44). Thus, we believed that an IV administration was the most appropriate method to investigate the vascular component of OIPN, since we expect it to cause a direct effect of oxaliplatin onto endothelial cells and other cellular components of the blood-nerve barrier. With that said, we have modified the aforementioned sentence to tone down our words.

For each figure, take care to explain each abbreviation of the figures (it is sometimes hard to understand their meaning).

We have modified the text and, above all, the figure legends accordingly to make them easier to understand.

Could you explain the choice of each drug dose?

Regarding the dose used for oxaliplatin (see response to Referee 1, point 1), we based our initial protocol/dose (three injections of 10 mg/kg) on the study by Ogihara et al. (J Pharmacol Sci 2019 141(4):131-8), which invariably showed significant OIPN hypersensitivity symptoms associated with peripheral vascular impairment. Thanks to referee 1's comment, we further tested different doses of oxaliplatin, 3 injections of 2, 5 and 10 mg/kg (i.e., a cumulative doses of 6, 15 and 30 mg/kg) and observed similar behavioral and molecular data at 5 and 10 mg/kg (but not at 2 mg/kg). To test the hypothesis that vasoconstriction is involved in the onset of OIPN symptoms in our model, we assessed the impact of previously described vasodilative drugs on the neuropathic symptoms. We used the recommended dose for tadalafil that has been tested in the context of pain (see Wang et al., PLoS One 2016 11(7):e0159665 & Ogihara et al., J Pharmacol Sci 2019 141(4):131-8), while we experimentally evaluated the impact of different doses of ambrisentan in order to identify the most relevant dose that resulted in robust and reproducible data. Since these two vasodilators act differently by either preventing vasoconstriction or enhancing vasodilation, we next decided to explore the combination of tadalafil and ambrisentan at previously established doses to determine whether a potentiation effect of the two drugs (Lachant et al., Am J Physiol Lung Cell Mol Physiol 2019 317(4):L445-L455) could be more efficient in preventing oxaliplatin-induced neuropathic symptoms.

Could you explain the choice of each gene/protein as biomarker of each structure/function

assessed? There are a lot of gene/protein assessed and it would help the reader to repeat some works)

We agree and have provided a table that includes all the markers and the cells/tissues they label. This has also been implemented in the text/figures whenever possible.

Using ambrisentan and/or tadalafil, would it be possible that these drugs increase the neurotoxicity of oxaliplatin, by increased the blood flow in nerves, and consequently the nerve/neuron exposures?

We do not think that vasodilators increase oxaliplatin neurotoxicity, since we observed a significant improvement of the neuropathic symptoms as assayed by the behavioral tests with both short- (Fig 6) or long-term administration of oxaliplatin and tadalafil (preliminary data, see Figure below).

Figure Legend: paw withdrawal threshold before and after injections of oxaliplatin (Oxa), oxaliplatin and tadalafil (Oxa TADA) or control (Sham). Oxaliplatin was intravenously injected once a week (10 mg/kg) and tadalafil orally dispensed every other day. n = 8 to 14 mice, Mann-Whitney test, two-tailed: *p<0.05, ** p<0.01, *** p<0.001

Moreover, some alternative strategies to prevent OIPN propose to reduce the blood flow in extremities to prevent CIPN, such as cryotherapy (<https://pubmed.ncbi.nlm.nih.gov/32955997/>) or compression (<https://pubmed.ncbi.nlm.nih.gov/38060077/>) of extremities. Do you have some explanations of such opposite strategies in comparison to your results? Does ambrisentan and/or tadalafil may prevent chronic OIPN?

Our strategy to prevent OIPN is different but not opposed to those described in these cited papers. Indeed, the objective of cryotherapy or compression is to induce anesthesia of the intra-epidermal nerve fibers by limiting blood flow. These two approaches thus cause the inhibition of the nociceptors, while our strategy aims at restoring normoxia in the peripheral nerves to prevent the occurrence of neuropathic symptoms, supposedly without inhibiting nociception (see answer to the last point below). Furthermore, our preliminary analysis has shown that tadalafil also protects against chronic oxaliplatin administration (cumulative dose of 60 mg/kg for more than 5 weeks, see Figure above). Finally, vasoconstriction at the periphery is thought to prevent IENF or individual neuronal fibers exposure to oxaliplatin that could be

detrimental, while nerve homeostasis is maintained by vasodilation to maintain proper perfusion and normoxia.

In some way, the first part of the work on transporters is not very useful for the final results. However, these explorations rise the question of a possible interaction between ambrisentan and/or tadalafil and oxaliplatin (blocking transporters involved in the oxaliplatin influx or efflux)? Is there any assessment of platinum exposure of DRG and nerves after ambrisentan and/or tadalafil treatment?

To address this point, we undertook the analysis of platinum exposure using mass spectrometry, which showed that tadalafil did not interfere with the accumulation of platinum in the plasma, DRGs and sciatic nerves of oxaliplatin-treated animals. These results have been added to Fig 6 (see extract below). We thereby believe that tadalafil does not hamper the activity of oxaliplatin transporters nor its accumulation.

(From Fig 6)

Finally, may ambrisentan and/or tadalafil have analgesic effect in control animals? Or do you have some arguments that it is not the case?

This is an intriguing point although according to the public drug databases and the French and British drug reference sites (<https://www.vidal.fr> & <https://www.nhs.uk>), tadalafil and ambrisentan are not described as presenting with an analgesic action. On the contrary, one of tadalafil known side effects, which affects close to 10% of treated patients, is myalgia and back pain (<https://www.nhs.uk> & Seftel et al., *Int J Impot Res* 2005 17(5):455-61). Yet, a few studies identified an antinociceptive/analgesic effect of tadalafil in pain models (Mehanna et al., *Toxicol Appl Pharmacol* 2018 352:170-5) or in a rat model of arthritis (Rocha et al., *Br J Pharmacol* 2011 164(2b):828-35). Likewise, an analog of Ambrisentan although antagonizing both endothelin receptors (ET_A and ET_B), Bosentan, showed an antinociceptive and anti-inflammatory activity in a mouse model of arthritis (Imhof et al., *Arthritis Res & Ther* 2011 13:R97). However, the latter study showed that the antinociceptive effects of bosentan are predominantly mediated via the ET_B receptor, while ambrisentan is a selective antagonist of ET_A . In view of these articles, we cannot exclude with certainty that tadalafil shows analgesic properties, although if an antinociceptive effect of tadalafil may explain the behavioral rescue of our oxaliplatin-treated mouse neuropathic pain, how could it be correlated with the rescue of hypoxia-associated molecular markers, such as HIF-1 α or downstream VEGFR-1 signaling?

November 1, 2024

RE: Life Science Alliance Manuscript #LSA-2024-02791-TR

Dr. Isabelle Brunet
Centre Interdisciplinaire de Recherche en Biologie
11 Place Marcelin Berthelot
Paris 75005
France

Dear Dr. Brunet,

Thank you for submitting your revised manuscript entitled "Vascular dysfunction is at the onset of oxaliplatin-induced peripheral neuropathy symptoms in mice". We would be happy to publish your paper in Life Science Alliance pending final revisions necessary to meet our formatting guidelines.

- please be sure that the authorship listing and order is correct
- please upload your main manuscript text as an editable doc file
- please upload your main and supplementary figures as single files
- please add the Twitter handle of your host institute/organization as well as your own or/and one of the authors in our system
- please be sure that the authorship listing and order is correct
- please remove figures from the manuscript file and upload them separately
- please add your main, supplementary figure, and table legends to the main manuscript text after the references section
- figures S2 and S5 have only one panel...please remove label A from the figures and their legends
- please upload your Tables in editable .doc or excel format
- please use the [10 author names et al.] format in your references (i.e., limit the author names to the first 10)
- please add callouts for Figures 8A, B; S1A, B; S3A-C; S4A-E; S6A, B, and Table 1 and 2 to your main manuscript text

FIGURE CHECKS:

- please add sizes next to blots in Figure 7C and E

A. FINAL FILES:

B. MANUSCRIPT ORGANIZATION AND FORMATTING:

Sincerely,

November 12, 2024

RE: Life Science Alliance Manuscript #LSA-2024-02791-TRR

Dr. Isabelle Brunet
Centre Interdisciplinaire de Recherche en Biologie
11 Place Marcelin Berthelot
Paris 75005
France

Dear Dr. Brunet,

Thank you for submitting your Research Article entitled "Vascular dysfunction is at the onset of oxaliplatin-induced peripheral neuropathy symptoms in mice". It is a pleasure to let you know that your manuscript is now accepted for publication in Life Science Alliance. Congratulations on this interesting work.

DISTRIBUTION OF MATERIALS:

Again, congratulations on a very nice paper. I hope you found the review process to be constructive and are pleased with how the manuscript was handled editorially. We look forward to future exciting submissions from your lab.

Sincerely,
